# Flexible Antennas: A Review

**DOI:** 10.3390/mi11090847

**Published:** 2020-09-11

**Authors:** Sharadindu Gopal Kirtania, Alan Wesley Elger, Md. Rabiul Hasan, Anna Wisniewska, Karthik Sekhar, Tutku Karacolak, Praveen Kumar Sekhar

**Affiliations:** 1School of Engineering and Computer Science, Washington State University Vancouver, Vancouver, WA 98686, USA; sharadindu.kirtania@wsu.edu (S.G.K.); alan.elger@wsu.edu (A.W.E.); mdrabiul.hasan@wsu.edu (M.R.H.); anna.wisniewska@wsu.edu (A.W.); tutku.karacolak@wsu.edu (T.K.); 2Department of ECE, Faculty of Engineering and Technology, SRM Institute of Science and Technology, Vadapalani Campus, No.1, Jawaharlal Nehru Road, Vadapalani, Chennai, TN 600026, India; karthiks1@srmist.edu.in

**Keywords:** flexible antenna, wearable antenna, 3-D printing, specific absorption rate (SAR), Internet of Things (IoT), implantable antennas, ingestible antennas, bending analysis

## Abstract

The field of flexible antennas is witnessing an exponential growth due to the demand for wearable devices, Internet of Things (IoT) framework, point of care devices, personalized medicine platform, 5G technology, wireless sensor networks, and communication devices with a smaller form factor to name a few. The choice of non-rigid antennas is application specific and depends on the type of substrate, materials used, processing techniques, antenna performance, and the surrounding environment. There are numerous design innovations, new materials and material properties, intriguing fabrication methods, and niche applications. This review article focuses on the need for flexible antennas, materials, and processes used for fabricating the antennas, various material properties influencing antenna performance, and specific biomedical applications accompanied by the design considerations. After a comprehensive treatment of the above-mentioned topics, the article will focus on inherent challenges and future prospects of flexible antennas. Finally, an insight into the application of flexible antenna on future wireless solutions is discussed.

## 1. Introduction

The availability of high speed, massive capacity, and low latency 5G network has enabled the ‘Fourth Industrial Revolution’ [1]. Every sector will benefit from 5G networks ranging from 3-D imaging, advanced health care, streaming services, and smart cities, to name a few [2]. Further, a strong 5G network is essential to the proper functioning of the Internet of Things (IoT) devices [3,4]. A visual representation of the interconnection between IoT things and the 5G network is shown in Figure 1.

One of the critical barriers to technological advancements of next-generation IoT related devices is inflexibility stemming from form factor and weight considerations. While there have been orders of magnitudes of advances in miniaturization, flexibility is a feature that is hard to conquer. Recent innovations in engineered materials have been leveraged to augment the field of flexible electronics. Flexible electronic devices are often lightweight, portable, less expensive, environment friendly, and disposable [6]. The flexible electronics market is expected to reach 40.37 billion in revenue by 2023 [7,8]. Figure 2 shows the various applications of flexible electronic devices. Flexible electronic systems require the integration of flexible antennas operating in specific frequency bands to provide wireless connectivity, which is a necessity in today’s information-oriented society.

The markets for flexible wireless devices are rapidly increasing partly due to the demands in wearable and implantable devices for health-monitoring systems and daily-life wireless devices (e.g., cell phones, laptop computers, etc.). For this reason, the need for flexible printed antennas has increased in recent years, especially for biomedical applications [10,11]. Specifically, flexible antennas are a major component in the implementation of in vivo monitoring of vital signs, regulation of organ functions, neural interfaces, continuous gait analysis, intracranial sensors, drug delivery systems, and countless other functions [12,13]. To integrate the devices onto the human body characterized by curvilinear surfaces and dynamically changing motions, the device must be conformal and physically flexible or even stretchable. Because the bending stiffness of a thin film structure that characterizes its resistance against bending deformation roughly scales with the cubic of its thickness thinning down the thickness of the structure represents an effective means to enable flexible/bendable antennas.

Apart from biomedical applications, there is shared interest between Federal agencies, industry, and academia in developing a flexible antenna for extreme conditions. Federal agencies’ applications for high-temperature flexible devices include: the new gas-cooled nuclear reactor (high-temperature H_2_ safety monitors) and strict automotive emissions control requirements (tailpipe sensors) from the Department of Energy (DOE); requirement of flexible antennas and antenna radomes with extreme thermal shock resistance for missile applications and high-temperature substrates for hypersonic antennas for the Department of Defense (DoD); communication solution for beyond-line-of-sight communications on small- to medium-scale unmanned aircraft systems (UAS) for the National Aeronautics and Space Administration (NASA); and Wireless Physiological and Environmental Monitoring (WiPEM) system requirement (for first responders) from the Department of Homeland Security (DHS).

Several recent investigations have reported that provide extensive review on materials, fabrication, and applications of different flexible and wearable antennas [14,15,16,17,18,19]. A practical guideline for designing and fabricating both nontextile and fully textile wearable antennas has been reported in [14]. In another article [15], the main focus was the different fabrication technologies for flexible antennas along with a design and analysis of a printed monopole antenna. A complete survey of recently used materials and fabrication methods of wearable antennas, ranging from Very High Frequency (VHF) to millimeter-wave band was presented earlier [16]. In another article [17], the main focus of the review was on materials and fabrication techniques for only wearable antennas designs with their applications, limitations, and solutions. The recent progress in the field of wearable ultrawideband antennas and their application in wireless body area network (WBAN) systems were demonstrated in an earlier study [18]. Different types of implantable antennas, their design requirements, and performance comparison was surveyed in a prior article [19]. The uniqueness of this article stems from the fact that this paper is the fact that we cover the entire area of flexible antenna rather than focusing on any subset such as wearables. Further, the authors discuss antenna applications covering a wide frequency bands unlike other articles restricting the scope on the working frequency. In addition, the original of this article is the focus on the investigations in the last 5 years. 

In any wireless application, the choice and design of the antenna vary depending on the environment, transmission strength, and frequency range [20]. Further, the performance of the antenna depends on the material used, the type of fabrication technique employed, and the substrate properties. In this context, the article reviews the research trend in conductive materials used, the substrates, the different fabrication techniques to realize the flexible antenna, and their diverse applications. Besides, the challenges in flexible antenna research and future directions are highlighted. 

## 2. Materials for Flexible Antennas

Flexible antennas are fabricated using various conductive materials and substrates. The substrate is chosen based on their dielectric properties, tolerance to mechanical deformations (bending, twisting, and wrapping), susceptibility to miniaturization, and endurance in the external environment. In contrast, the selection of conductive material (based on electrical conductivity) dictates the antenna performance, such as radiation efficiency. 

### 2.1. Conductive Materials

In wireless applications, the realization of conductive patterns with superior electrical conductivity is essential for ensuring high gain, efficiency, and bandwidth. Additionally, resistance to degradation due to mechanical deformation is another desired feature for the conductive material. Nanoparticle (NP) inks (i.e., silver and copper) are often preferred for fabricating flexible antennas due to their high electrical conductivity. Silver-nanoparticle ink edges over copper nanoparticles due to their low rate of oxide formation [21]. Very few investigations have been observed for flexible antennas based on copper-based nanoparticles [22]. Besides nanoparticles, electro-textile materials like Ni/Ag-plated, Flectron (copper-coated nylon fabric), and nonwoven conductive fabrics (NWCFs) are generally used in flexible antennas. Various types of textile and non-textile conductive materials for developing flexible antennas have been reviewed in an earlier article [14]. Adhesive copper [14], copper tapes [23], and copper cladding [24] have been reported in the development of flexible antennas.

Figure 3 shows the images of antennas fabricating with different conducting materials. Conductive polymers like polyaniline (PANI) [25], polypyrrole (PPy) [26], and poly(3,4-ethylenedioxythiophene) polystyrene sulfonate (PEDOT:PSS) [27] seem to be promising materials for flexible and wearable antennas. The low conductivity of conductive polymers was improved by adding carbon nanotubes [28], graphene [29], and carbon nanoparticles [30] (Figure 3). Flexible antennas using graphene are promising due to their decent electrical conductivity and excellent mechanical properties. Graphene paper [31], graphene nanoflake ink [32,33], graphene oxide ink [34], and graphene nanoparticle ink [35] have been used in prior studies for fabricating flexible antennas. The performance of flexible antennas relies heavily on the fabricated conducting traces with high deformation sustainability while maintaining electrical conductivity [36]. For accommodating mechanical strain and deformation without deteriorating the performance of the antennas, different stretchable conductive materials exploit doping to improve their conductivity. Some of the examples include silver nanowire embedded silicone [37], silver loaded fluorine rubber [38], carbon nanotubes (CNT)-based conductive polymers [28,39], liquid metals in the stretchable substrate [40], and use of stretchable fabric itself [41]. Table 1 lists the different conductive materials used in the fabrication of a flexible antenna along with their conductivity values. 

### 2.2. Substrates

The substrate material used in flexible antenna needs to possess minimal dielectric loss, low relative permittivity, low coefficient of thermal expansion, and high thermal conductivity [50]. Such a constraint is driven by the need for increased efficiency (in different environments) at the cost of larger antenna size. An exception to the above-mentioned fact is the need for large dielectric constant for miniaturized antennas. Three types of substrates have often surfaced in the fabrication of flexible antennas: thin glass, metal foils, and plastics or polymer substrates [51]. Though thin glass is bendable, the intrinsic brittle property restricts its utility. Metal foils can sustain high temperatures and provide inorganic materials to be deposited on it, but the surface roughness and high cost of the materials limit its applications [52]. Plastic or polymer materials are the best candidates for flexible antenna applications which include: (1) the thermoplastic semicrystalline polymers: polyethylene terephthalate (PET) and polyethylene naphthalate (PEN), (2) the thermoplastic noncrystalline polymers: polycarbonate (PC) and polyethersulphone (PES), and (3) high-glass transition temperature, *T*g materials: polyimide (PI) [53].

They are popular and attractive in recent years for flexible electronics due to their robustness, flexibility, wettability, and stretchability. Due to high *T*g, polyimide is one of the most preferred materials for flexible antennas, which had been used as a substrate in prior studies [54,55,56,57,58,59,60,61,62]. Sanusi et al. [55] reported on the design and performance of an artificial magnetic conductor (AMC)-backed dipole antenna on Kapton Polymiode for RF energy harvesting in the context of next-generation blood irradiation systems. PET and PEN are preferred in many flexible antenna designs due to excellent electrical, mechanical, and moisture resistant properties [63]. For fabricating different types of flexible antennas, PET has been used routinely due to its excellent conformal behavior and mechanical stability [63,64,65,66,67]. In an earlier report [68], an inkjet-printed slotted disc monopole antenna was designed, printed, and analyzed at 2.45 GHz industrial, scientific and medical (ISM) band on PET for early detection of brain stroke. Next, a new all-organic flexible patch antenna was fabricated with PEDOT: PSS on a selected PET fabric substrate for next-generation wearable antennas [69]. A coplanar square monopole operating at 60 GHz with 68% total efficiency and 1.86 dBi maximum realized gain was presented in [70] over a PEN substrate. PET and PEN substrate have excellent conformality, but low glass transition temperature limits their application for the high-temperature condition.

Flexible antennas made for wearable purposes need unique attributes such as limited visibility for the user, robust antenna performance in different conditions, mechanical stability, and withstanding rigor, such as washing and ironing [71]. Different types of substrates used in wearable antennas have been reported in a prior article [3]. Felt, fleece, silk, and Cardura, off-the-shelf (electro) textile materials, and standard apparel are a few examples of substrates that have been used for wearable/flexible antennas. The use of polydimethylsiloxane (PDMS) polymer as a substrate has been emerging because of its low Young’s modulus (<3 MPa) suggesting high flexibility/conformality [72]. However, the development of a flexible antenna is limited on the PDMS substrate due to the weak metal–polymer adhesion. Nevertheless, some solutions to this issue have been found in literature such as implanting carbon nanotube sheets [73] or different microspheres like glass, phenol or silicate [73] or nanowires (AgNWs) [37,74], injecting liquid metal [75], and oxygen plasma treatment on the PDMS surface [76,77,78].

Paper substrate has been preferred for flexible antennas due to the cost-effective nature and ease of manufacturing. A coplanar waveguide (CPW)-fed flexible UWB antenna operating from 3.2–30 GHz (161% fractional bandwidth (FBW)) for IoT applications on photo paper has been presented in [79]. Ullah et al. [80] have demonstrated a paper substrate-based flexible antenna design for intrabody telemedicine systems in the 2.4 GHz industrial, scientific, and medical radio (ISM) bands. Liquid crystal polymer (LCP) is a flexible-printed-circuit like thin-film substrate and regarded as attractive for high-frequency flexible antennas due to low dielectric loss, lower moisture absorption, resistant to chemicals, and can withstand temperatures up to 300 °C [81]. A flexible millimeter-wave (mm-wave) antenna array on a thin film of flexible liquid crystal polymer for the fifth-generation (5G) wireless networks operating at Ka-band (26.5–40 GHz) was presented in [82] and a dual sensed liquid crystal polymer-based metamaterial loaded circularly polarized flexible antenna on LCP substrate is proposed in [83] working in Worldwide Interoperability for Microwave Access (WiMAX) and wireless Local Area Network (WLAN) band. Table 2 shows commonly used substrates for flexible antenna fabrication along with their dielectric constant, dielectric loss, and thickness values. Figure 4 shows flexible antenna prototypes on different substrates. 

It can be concluded without a doubt the choice of substrate material is of paramount importance in the realization of flexible antennas. Due to their conformal behavior and operational suitability, flexible materials have gained immense interest. These flexible materials need to be chosen carefully to withstand the physical deformation conditions such as bending, stretching, and even twisting while maintaining its functionality. Flexible antennas require low-loss dielectric materials as their substrate and highly conductive materials as conductors for efficient EM radiation reception/transmission. The recent flexible substrates being introduced for wearable/flexible antennas include the likes of Kapton, PET, paper, liquid crystal polymer, different fabrics, and paper due to their unique physical, electrical, and mechanical properties.

## 3. Fabrication Techniques for Flexible Antennas

The performance of a flexible antenna is determined by the fabrication method (which is different for different substrates). The common fabrication techniques include wet-etching, inkjet printing, screen printing, and other special methods for fabricating flexible wearable antennas. A detailed overview of different fabrication techniques of flexible antennas can be found in earlier reports [15,16].

### 3.1. Inkjet Printing

Inkjet-printing technology has emerged as an alternative to conventional fabrication techniques such as etching and milling. It is an additive process so that the design is directly transferred on to the substrate without any masks and ensures less material wastage [90]. It is the preferred fabrication technique for polymeric substrates like polyimide, PET, paper due to its accurate and speedy prototyping fabrication method [90]. Figure 5 shows the overview of the printing process and the printer assembly.

For printing purposes, nanoparticle metallic inks, graphene nanoflake inks, metal-organic inks are utilized. The printing technique can be sorted into two types: drop-on-demand (DoD) and continuous inkjet. Drop on-demand print heads apply pressurized pulses to ink with either a piezo or thermoelement in which drives a drop from a nozzle when needed [15]. New generation printers offer precise printing using picolitre volume cartridges. Printing quality is controlled by the jetting waveform, the jetting voltage of the nozzles, the jetting frequency, the cartridge temperature, the platen temperature (where the substrate is placed), and the resolution of the pattern [91,92]. After the printing of the antenna design, sintering is necessary for removing the solvent and capping agent and attaining electrical conductivity [93].

An example of inkjet printing is the use of silver nanoparticle ink to fabricate a wideband right-hand circularly polarized high-gain 4 × 4 microstrip patch array antenna on a PET substrate using Epson stylus c88 series printer [65]. Another example of an inkjet-printed antenna is epidermal antennas suitable for radio-frequency identification (RFID), and sensing on transparent PET film has been presented in an earlier study [66]. The unique feature of this antenna fabrication method is that no heat sintering is needed (printer model is Brother MFC-J5910DW). A miniaturized fully inkjet-printed flexible multiple-input-multiple-output (MIMO) antenna for ultrawideband (UWB) application was proposed on Kapton polyimide substrate using a Dimatix DMP 2800 printer [59]. Using Dimatix DMP 2831 printer, a flexible, wearable, and reversibly deformable CPW fed antenna was designed on PET substrate using silver nanoparticles [94]. A high-gain, multidirector Yagi-Uda antennas for use within the 2.45-GHz ISM band were realized using silver and dielectric ink on LCP substrate using the same printer [95].

Another work using photo paper to fabricate reconfigurable multiband antenna via two p-i-n diodes was reported [96]. A new silver nanoparticle-based conductive ink, having a built-in sintering mechanism, which is triggered during the drying of the printed pattern has been used to reduce the equipment cost and eliminate the complexity of post-treatment of the printed layer [97]. Meanwhile, chemically cured conductive ink (Ag nanoparticle) was used for fabricating a Z-shape antenna for operation in the ISM band (2.45 GHz) on a low-cost pre-treated PET substrate [98]. Printing resolution for inkjet-printed flexible antenna depends on the surface roughness of the substrate. For smooth substrates like Polyimide, PET, PEN, LCP, photo paper, etc., an excellent pattern resolution is achievable. For wearable, flexible substrates like E-textiles having the weaving of warp and weft yarns typically have an uneven surface. Hence, adequate resolution remains a challenge [99]. For fabricating wearable/flexible antenna, stitching, embroidery, and substrate integrated waveguide base (SIW) methods are mostly followed.

### 3.2. Screen Printing

Screen printing is a simple, fast, cost-effective, and viable solution for fabricating flexible electronics, which has been widely adopted to implement RFID antennas by printing conductive inks or pastes onto low-cost, flexible substrates such as PET, paper, and textile substrates [100]. It is a woven screen-based technique having different thicknesses and thread densities. A squeegee blade is driven down, forcing the screen into contact with the substrate to produce a printed pattern. Thus, the desired pattern is formed by the ink ejecting through the exposed areas of the screen on the affixed substrate [101]. It is also an additive process like inkjet printing as opposed to the subtractive process of chemical etching, which makes it more cost-effective and environmentally friendly. A screen-printed dual-polarization 2.45 GHz antenna and rectenna on polycotton for RF power transfer and harvesting were demonstrated in a prior article [102]. The rectenna was tested and compared with a similar FR4 rectenna, and the performance was found a third of standard FR4.

A high-frequency RFID reader antenna system was fabricated by screen printing silver (Ag) particle inks onto a flexible paper substrate operating at 13.56 MHz [103]. This screen printed antenna’s performance (Q factor) was significantly lower compared to a Cu made loop antenna element with the same geometry. For improving this, antenna DC resistance needs to be reduced by adding more silver printing layers, which eventually increases the manufacturing cost. As screen printing is cost-effective, a partially reflective surface with a parasitic patch array to create a simple beam-switching, low-profile, and flexible wearable detection system was designed on a flexible substrate from PremixGroup [104]. The antenna array was printed using Aurel Automation 900 screen printer, and the antenna was optimized for the 77 GHz band and had a high gain of 11.2 dB, which can detect objects within a range of 10 m. The fabrication process has been given in Figure 6. A DEK Horizon 03i (ASM Assembly Systems Weymouth Limited, Weymouth, UK) semiautomatic screen printer was used to fabricate a graphene-flakes-based wideband elliptical dipole antenna on a polyimide substrate operating from 2 to 5 GHz for low-cost wireless communications applications [105]. Screen printing is cost-effective compared to other fabrication technologies of flexible antennas. However, it has some limitations like resolution dependence on the surface quality of the substrates, the limited layer control, and lack of thickness control for the conductive layer. Figure 6 shows the fabrication of a flexible antenna using the screen printing process and a sample prototype.

### 3.3. 3-D Printing

Recently, additive 3-D printing techniques for flexible antennas are gaining popularity with a myriad of commercially available printing materials and processes. It exhibits several advantages like in-house-made, fast fabrication of complex 3-D structures with various materials, and the capability to change the density of the printed object [106,107,108]. The flexibility to realize complex 3-D shapes from bulk materials and 3-D printing of flexible materials like polymers, metals, ceramics, and even biological tissues make it attractive for antenna design [109]. Polymers, such as thermosets and thermoplastics, are used as 3-D printing materials for flexible antenna applications. The common printing techniques of the polymers are Fused Deposition Modeling (FDM), Stereolithography (SLA), Direct Light Processing (DLP), and Material Jetting (MJ) [110]. The most common 3-D printing technology is FDM. In FDM, the filament is fed through to the extrusion head of the printer, and the motor of the heated-nozzle drives the filament melting it. The printer then lays down the melted material at a precise location, where it cools down and solidifies. The process repeats by stacking up the part layer-by-layer [110].

One of the first examples of the exploitation of 3-D-printing in microwave components and antennas fabrication was presented in an earlier article [107]. NinjaFlex, a new 3-D printable, flexible filament, has been adopted for manufacturing a 3-D printed patch antenna. FDM technique has been used to realize 3-D printing substrate. A linearly polarized patch antenna was designed and implemented on the NinjaFlex substrate 100% infill at 2.4 GHz operating frequency (Figure 7a). Experimental verification under nominal and bending conditions showed good agreement with the simulation. A ‘button-shaped’ compact RFID tag fabricated by the combination of 3-D printing and inkjet printing technologies was reported in [111] for wearable applications (Figure 7c). The antenna showed a good performance with a measured maximum reading range of 2.1 m in the RFID Federal Communications Commission band (902–928 MHz). A proof-of-concept of the fabrication and performance analysis of a flexible and stretchable wearable antenna on a 3-D printed NinjaFlex substrate was presented in [112].

The radiator of the antenna was brush-painted from the stretchable silver conductive paste (Figure 7b). The antenna’s wireless performance under flat and bending conditions was satisfactory. Specific absorption rate (SAR) simulations validate its use for wearable applications. The antenna showed impedance bandwidth of 990 MHz (1.94–2.93 GHz) with a peak gain of −7.2 dB at 2.45 GHz. A bow-tie antenna with a CPW feed structure was fabricated using a desktop 3-D printer [113]. Polylactic Acid (PLA) and Acrylonitrile butadiene styrene (ABS) filaments were used as the dielectric and conductive parts of the antenna, respectively, which showed wide bandwidth, flexible structure, lightweight, and small size. Recently, a 3-D flexible, miniaturized inverted-F antenna for wearable applications was designed, manufactured using the Galinstan liquid metal to realize the radiating element, the NinjaFlex flexible plastic to realize dielectric substrate through a 3-D FDM printing process (Figure 7d) and the electro-textile copper constituting the antenna ground plane [114]. The performance of the antenna in several bent configurations and in the presence of the human body was found satisfactory.

### 3.4. Chemical Etching

Chemical etching, often accompanied by photolithography, emerged in the 1960s as a branch-out of the Printed Circuit Board (PCB) industry is the process of fabricating metallic patterns using photoresist and etchants to mill out a selected area corrosively. For fabricating complex designs with high resolution accurately, it is the best choice among all other fabrication techniques [115]. Organic polymers are suitable for photoresists as their chemical characteristics change when they are exposed to ultraviolet light. Current practice in the photolithography based antenna and RF circuits industry relies mainly on positive resists since they present higher resolution than negative resists. A multilayer type of flexible monopole antenna was designed and fabricated on a transparent polyimide substrate for application in wearable glasses in an earlier article [116]. A 100-nm-thick indium–zinc–tin oxide (IZTO)/Ag/IZTO (IAI) is a transparent (81.1%) conducting oxide electrode, which was used as the conductors of antennas and ground planes of the wearable glasses. Physical Vapor Deposition (PVD) process is employed to fabricate this multilayer type of flexible antennas. The fabrication processes are shown in Figure 8.

This work shows the feasibility of wearable, flexible antennas for optical and electrical applications using the photolithography process. A practical 5.8 GHz antenna for the wireless operation was fabricated on a flexible glass by photolithography using Shipley S1813 photoresist and a Heidelberg Instruments µPG 101 pattern generator. [117]. A 100 nm thick aluminum-doped $SiO_{2}$ layer served as a buffer layer on the glass before 650 nm thick sputter deposition of the indium–tin oxide (ITO) layer. After lithography, the ITO was etched using an oxalic acid etch was annealed in a rapid thermal annealing system in nitrogen at 500 ℃ for 5 min with a ramp rate of 5 C/s. Loops and grid configurations are possible using this methodology. A small size (2.5 cm × 5 cm) epidermal RFID antenna was fabricated using four different fabrication systems, including photolithography [118]. The process is shown in Figure 8. Here, Au is used as the antenna conductor with a Ti/W adhesion layer on the polyimide substrate. Though high-resolution throughput has been possible using microfabrication, high costs associated with cleanroom facilities, photo masks, photolithography chemicals, and human resources will prevent the epidermal antenna from being inexpensive and disposable. A process flow using photolithography and sputtering was studied for high-quality Cu antenna fabrication on the PET substrate without any damage to the supporting layer [119]. To conclude, complex and fine detailed antennas can be fabricated using photolithography. However, the lengthy process, along with the involvement of dangerous chemicals, high-end expensive cleanroom equipment, photomasks, and chemicals with human resources, limit its application in fabricating flexible antennas.

### 3.5. Special Fabrication Techniques for Flexible Wearable Antennas

The special fabrication techniques for fabricating flexible antenna can be divided into the following categories: (1) SIW based technology; (2) stitching and embroidery; (3) the use of conductive textile yarns to embroider the conductive patterns of the antenna on a non-conductive textile substrate; and (4) inkjet and screen printed printing on non-conductive textile materials. This article focuses on the first two techniques.

#### 3.5.1. Substrate Integrated Waveguide (SIW) Based Technology

A relatively new method referred to as Substrate Integrated Waveguide (SIW) is highly desirable to realize future system on substrate (SoS) platforms for developing high-performance mm-wave systems [119]. This structure ensures the confinement of electric fields inside the cavity by the use of shorting vias on its sidewalls, backed by the full ground plane. The main advantages of the SIW based technology are the improvement of the Q-factor of the antenna and improving isolation between the antenna and wearers body. A compact substrate integrated waveguide (SIW) based wearable tri-band (ISM, WiMAX, and Military application) leather antenna, designed for optimal on-body performance was proposed in [120]. By using brass eyelets and a combination of conducting and non-conductive leather materials, a substrate integrated waveguide cavity with a ground plane was realized, and miniaturization achieved by introducing slits. In another study [121], a low-cost SIW based antenna made of pure copper taffeta fabric etched on a woolen felt substrate operating from 2.27 to 3.61 GHz was proposed (Figure 9b).

The antenna topology is based on a folded cavity with an annular ring as a radiating element with a 73% radiation efficiency. Next, a conductive fabric-based SIW based antenna was fabricated on a woolen substrate for off-body applications in an earlier report [122] (Figure 9a). Antennas fabricated using the SIW method is shown in Figure 9. A novel fleece fabric WLAN antenna for wearable applications such as sportswear and emergency worker outfits was reported previously [123].

#### 3.5.2. Stitching and Embroidery

Weaving or knitting the conductive textile on the substrate is another method for fabricating flexible wearable antennas. Knitting copper on a fleece substrate, the first compact fabric antenna design for commercial smart clothing was presented in an earlier article [124]. Embroidery with conductive yarn is a simple fabrication method with great potential in flexible wearable antenna fabrication due to its compatibility with nonelectronic textile processing capabilities. An embroidered antenna-IC in–erconnections in passive UHF RFID electrotextile tags and the possibility of creating a planar dipole tag by only embroidering the borderlines of the full antenna shape was studied earlier [125]. A 160-mm-diameter Archimedean spiral antenna was weaved using seven-filament silver-plated copper Elektrisola E-threads on a Kevlar fabric substrate [126]. The application area of this antenna covered several wideband, conformal, and load-bearing applications, such as airborne and wearables. Recently, a mix of these two methods (stitching and embroidery) was presented in a study [127] (Figure 10c).

This work demonstrated the possibility of implementing an all-textile antenna, reducing the backward radiation via the use of a SIW topology. A novel embroidered metamaterial antenna based on as split ring resonator (SRR) electromagnetic bandgap (EBG) shielding structure was presented in [128] (Figure 10a). Split ring resonator (SRR) has been introduced in the antenna to reduce the SAR value, which would improve the antenna performance. For efficient antenna design, the quality, strength, and flexibility of the conductive yarns, the accuracy of the embroidery machine, stitching density, and direction on the fabric are the main factors to consider [129]. Figure 10b shows the embroidery metamaterial manufacturing process.

## 4. Applications of Flexible Antennas under Different Frequency Bands

Even the most conservative projection for the growth of the internet of things (IoT) shows that the global IoT industry set to reach over USD 363.3 billion by the year 2025 [130]. A significant portion of this market includes health monitoring and clinic therapeutic devices, medical microwave radiometry, wearables, vehicular navigation systems, etc. Because of the nature of these applications, the antenna used should be flexible, conformal, and stretchable to comply with the curvilinear surfaces and dynamic motions. Besides civilian applications, it also plays a vital role in the military domain. Most of the military devices are connected to a large ad-hoc network. Military personnel are required to carry a large amount of equipment with different sensors and health monitoring devices. Therefore, flexible, and lightweight antennas are desirable in the military sector to reduce the burden of the soldier. In this article, flexible antenna applications are delineated into two categories: below 12 GHz and above 12 GHz.

### 4.1. Below 12 GHz

The development of flexible materials has paved the way for innovation in antenna designs and new applications that were not possible with rigid substrates. For flexible antenna applications below 12 GHz, RFID tag or smart card systems are typically designed using the flexible antennas at ultra-high frequency (UHF) band. The ultra-wideband applications of the flexible antenna cover WiMAX, WiFi, lower band of 5G [131], and one of the ISM radio bands. For flexible display devices working in the UHF band, a dipole antenna was reported on the Kapton polyimide substrate [132]. Kapton substrate ensures mechanical robustness and low dielectric loss for this antenna. In the UHF spectrum, antennas for smart cards and RFID tags dominate. Flexible RFID tags for non-invasive sensor applications like patient tracking in the medical system, internet of things (IoT) devices, childcare centers, humidity, and temperature sensing have been reported [133,134,135,136].

In an earlier article [137], a photo paper-based flexible inkjet-printed RFID tag antenna was reported for UHF applications. The antenna had an omnidirectional radiation pattern with 4.57 m coverage with a universal UHF band, 865–960 MHz. A temporary tissue type flexible RFID tag antenna was proposed in a prior study [133], which had a maximum range of 1.2 m. In another study [32], graphene nanoflakes printed flexible antenna was reported that was mainly a meandered line dipole antenna. It covered frequency from 984 to 1052 MHz with a radiation efficiency of 32% and a gain of 4 dBi. The new 3-D printing technology was used to design an RFID tag antenna in [138]. And in [111], a button-shaped RFID tag antenna was proposed combining 3-D and inkjet printing technology. This antenna showed a maximum reading range of 2.1 m. A flexible 3-D printed RFID tag antenna [139] was shown that achieved a maximum 10.6 m read range, and even in several stretching conditions, it covers more than 7.4 m.

Flexible antennas offer a promising solution for body-centric medical, consumer electronics, and military applications. For wearable applications, besides UHF bands, a 2.45 GHz frequency is used extensively for ISM applications. A low profile, lightweight, and robust antenna is preferred for this type of application. For intrabody telemedicine systems, a flexible photo paper based antenna was proposed that operates at 2.33–2.53 GHz [80]. In another study [140], a wearable textile logo antenna was proposed that was designed for military applications operating at the ISM band. In the literature, many other flexible antennas were reported for on-body applications that fall in the ISM frequency band [141,142,143,144,145].

Another frequently used antenna type is the ultra-wideband (UWB) antenna. In 2002, Federal Communication Commissions (FCC) defined the UWB spectrum ranging from 3.1 to 10.6 GHz to comply with the demand for higher data rate. The proliferation of body-centric communications, a subcategory of wireless body area networks (WBAN), has encouraged researchers to focus on flexible wideband and ultra-wideband antennas. UWB antennas have significant features such as small electrical size, low cost, low power spectral density, and high data rate [146], and have found extensive use. Because of the lower spectral density, the antenna is less prone to interference with other signals [147]. The UWB antenna made with the textile substrate can be used for on-body applications because it has minimal effect on the human body [148,149,150,151,152]. Paper-based inkjet-printed UWB antenna was introduced first in earlier studies [153,154]. Later various shapes of the conducting patch were designed to improve the efficiency of the antenna [153,154,155,156]. A compact, the high efficient polymer-based flexible antenna was proposed in [157]. In this research, the authors used sticky tape and PEDOT as the substrate and the conducting material, respectively. Many other polymer-based flexible UWB antennas were reported in the literature. These include liquid crystal polymer [158,159,160,161], polydimethylsiloxane (PDMS) [162,163,164], graphene-assembled film [165], artificial magnetic conductor (AMC) [166,167,168,169,170], PET [171,172,173], paper [174], and polyamide [175,176,177].

### 4.2. Above 12 GHz

According to Radio Society of Great Britain (RSGB), above 12 GHz, the Ku band starts, and these high-frequency bands are primarily used for radar, satellite communications, astronomical observations, radio astronomy, and microwave remote sensing. For remote sensing, radar, and future communication systems, dual-polarization microstrip antenna arrays were reported on LCP substrate operating at 14 and 35 GHz. Dual-polarization and dual-frequency ensures higher capacity data transfer [81]. A flexible, washable, and reusable UWB fully textile-based wearable antenna was designed and analyzed in earlier reports [178]. It maintained excellent efficiency from 3 to 20 GHz conductive for medical monitoring applications and smart garments. For example, flexible graphene antennas in single and array on polyimide substrate (Figure 11a) operating at 15 GHz produced large bandwidth to support higher-speed for 5G applications [179].

In a prior study [180], a flexible, transparent, and wideband mm-wave slotted monopole antenna was designed, fabricated, and tested by inkjet printing of custom made silver nanowire (Ag NW) ink. The antenna showed ultra-wide bandwidth up to 26 GHz (from 18 to 44 GHz), the high radiation efficiency of 55%, and a maximum gain of 1.45 dBi. A comparison between CPW fed monopole antennas printed on PET and Epson paper operating at 20 GHz was reported earlier [181]. The antennas were printed using CuO based ink using an inkjet printer. A Y-shaped transparent and flexible coplanar waveguide fed (CPW) antennas operating from 23 to 29.5 GHz, covering the necessary frequency bands for 5G wireless communications, was modeled using silver-coated polyester film (AgHT) transparent conductive material over PET substrate in [182]. Another work of high-frequency application using inkjet-printed technology is a proximity-fed patch antenna, designed for the 24-GHz ISM band, which was demonstrated in [183]. This antenna was printed using multilayer printed technology on the LCP substrate. Frequency reconfigurable antennas can operate at different frequency bands. In another study [184], a reconfigurable wearable Millimeter Wave (MMW) antenna has been introduced covering 20.7–36 GHz range in different switch configurations. The antenna was printed on an LCP substrate using inkjet-printed technology. The dielectric characteristics of PDMS have been in V- and W-bands have been tested and experimented in [185].

A micromachined microstrip patch antenna was designed in the 60 GHz band on the PDMS substrate for assessing and comparing this technology with the alternative one (Figure 11b). A flexible millimeter-wave antenna array exhibits a bandwidth of 26–40 GHz with a peak gain of 11.35 dBi at 35 GHz, and a consistent high gain profile of above 9 dBi in the complete Ka-band [15]. An electromagnetic bandgap (EBG) structured mm-wave MIMO antenna operating at 24 GHz (ISM band), suitable for wearable applications, was proposed in [186] on flexible Rogers substrate (Figure 11c). The antenna parameters were studied in free space as well as on a human phantom under bending. The proposed antenna is suitable for wearable applications at mm-wave range due to its simple geometry and excellent performance in bending and on-body worn scenarios.

## 5. Miniaturization of Flexible Antenna

The desire to connect all electronic devices into the IoT, has accelerated the need for integrated smaller flexible antennas. As a result, research investigations for small antennas have been increasing. The major challenge for researchers in this field is reducing the size of the antenna in order to integrate with miniaturized devices without compromising the antenna performance parameters such as impedance matching, gain, bandwidth, radiation pattern, and efficiency. Although, it is a difficult and daunting task, researchers have found a number of creative approaches for shrinking antenna size. In the literature, there are various techniques proposed to reduce the antenna dimensions. Here, the focus is on the methods used for lowering the form factor of the flexible antennas.

The applied methods can be mainly classified into three groups: material based miniaturization, topology-based miniaturization, and use of electromagnetic bandgap (EBG) structures. The first technique to reduce the flexible antenna dimension is utilizing high relative permittivity materials. The operating frequency relies on the dielectric environment of the antenna. The higher value of the dielectric constant (k), the smaller the size of the antenna. In an earlier study [187], the authors proposed a silver nanowire/nano paper composite based antenna that has a very high dielectric constant, k = 726.5 at 1.1 GHz. This k value was much higher than the typical flexible substrate used for flexible antenna such as polyethylene naphthalate (PEN) with k = 3.4, polyethylene terephthalate (PET) with k = 3.1, and polyimide (PI) with k = 3.4. According to this research, using the proposed nanowire composite as a substrate downsized the antenna dimension by about a half. Bending and thermal tests were also conducted, showing the suitability of the composite for reducing the dimension of the flexible antenna. The same technique is applied in many other studies [188,189,190]. In another study [191], locally filled high permittivity substrate was used to reduce the antenna size.

Antenna downsizing based on the change of topology is a popular approach. Altering the geometry, the current density distribution, and the electrical dimensions change the antenna properties. Optimization is needed to ensure the certain property of an antenna. There are different meandering lines used to increase the electrical length and miniaturize the flexible antenna. In an earlier study [192], asymmetric meander line was used to reduce the size of the flexible antenna and increase the gain. An earlier report [193] presented a compact dual-band flexible antenna that used a meander line to downsize the antenna and extract dual-band characteristics. The fractal antenna can provide characteristics like a larger antenna with a smaller dimension because of the efficient use of the area. By applying the Minkowski fractal geometry, miniaturization was achieved in a wearable electro textile antenna [194]. An ultra-thin flexible antenna that consisted of rectangular fractal patches with a stub was demonstrated in [195]. This rectangular fractal patch was shown to achieve 30% miniaturization compared to the traditional quadrilateral fractal patch. A defected ground plane was adopted as a method to control gain, radiation, and the dimension of an antenna. A compact wearable antenna with a double flexible substrate was designed with this concept [196]. Etching or printing slots on the flexible substrate is another way to manipulate the characteristics of the flexible antenna. [61,197,198] showed the compact, flexible antenna with slots in the patch. In another study [199], Sierpinski carpet fractal antenna on Hilbert slot pattern ground was introduced. Shorting posts and cutting slots are two other common techniques that have been used in many studies [200,201,202,203]. The space-filling curve (SFC) is a popular method to reduce antenna size [204]. In an earlier study [205], the authors combined these two miniaturization techniques with half mode substrate integrity waveguide cavity to reduce the size of the antenna further.

Newly developed electromagnetic bandgap (EBG) structures have received attention for their ability to reduce the physical structure of the antenna without compromising on the radiation efficiency. Artificial magnetic conductors (AMC) and high impedance surfaces (HIS) [206] were applied to design low profile antennas.

The first EBG-based wearable antenna was introduced earlier [207]. Even though it was designed as a wearable antenna, the antenna is not built on a flexible substrate. However, this study shows for the first time how to incorporate the EBG surface to reduce the antenna dimension. AMC based flexible M shaped antenna was proposed [208] (Figure 12c) for telemedicine applications that used polyimide Kapton flexible material as the substrate. The AMC structure helps to isolate the antenna radiation and human tissue; besides, it reduces the impedance mismatch caused by the permittivity of the user’s body. A textile wearable EBG antenna was developed in [209], where fleece fabric was used as the substrate. Here the EBG surface improved the bandwidth by almost 50%, and the reduction of the antenna size is about 30%. The authors also investigated the rigidity of the antenna in different bending conditions and its effect on the impedance bandwidth. With the success of this work, researchers extended the research on EBG integration in antenna to develop compact, high-performance antenna in the following works [210,211,212,213,214,215,216,217,218,219] (Figure 12a). Photonic bandgap (PBG) structures, another form of EBG, can prevent the propagation of a certain wavelength because of its periodic nature. PBG is a 3-D structure with stacked EBG layers. It is usually a combination of multilayer metallic and tripod array. Earlier work [220] shows the effect of PBG material on a conventional antenna system and the way to reduce the size of the antenna without compromising radiation efficiency, gain, and impedance bandwidth. The authors also demonstrated a new flexible antenna using the proposed PBG material. PBG periodic structure was used in the conformal antenna and array to suppress the surface wave propagation [221] (Figure 12b). It is shown to help to reduce the effect of radiation on the cylindrical curvature that is supposed to affect the resonance frequency. The gain and directivity of the antenna were improved by using PBG.

## 6. Flexible Antennas for Implantable Applications

In recent years, the continuous advancing and revolutionizing of health care systems towards the advancement of an efficient system to increase the quality of life as well as implementing future IoT in medical sector. An implantable antenna system transmits and stores the recorded physiological parameters, conditions for real-time communication. So, flexible antennas play a huge role in implantable antenna applications and they are receiving significant attention to the researchers and thus have become a current research focus. Flexible antennas are quite necessary as most of them are from polymeric substrate which can be biocompatible in nature.

For designing an implantable antenna, the basic requirements are the small size along with proper placement inside the human body, larger bandwidth, flexibility, and low specific absorption rate (SAR). It is also challenging due to the different dielectric constant of various tissues and organs of the human body. A flexible folded slot dipole antenna embedded in PDMS for implantation into the human body was pursued in a prior study [222] (Figure 13a).

The antenna performance and SAR measurements were done using a liquid mimicking the human muscle tissue varying dielectric nominal values mimicking different tissues.

The EM characteristics of the antenna were found stable for various properties of the surrounding tissues. Flexibility tests were done by bending the antenna over two different radii curvatures, which shifted the resonance frequency slightly within the bandwidth. An implantable wideband low SAR antenna on a flexible PDMS substrate was proposed earlier [223] (Figure 13b). The unique design feature of the antennas was responsible for achieving low SAR. Pork loin and muscle mimicking gels were used for experimenting with the antenna features. Bending analysis of the antenna showed a slight variation in frequency from the flat condition.

An implantable ring-slot antenna in the ISM frequency band was proposed using the grounded metamaterial technique in [61]. This antenna was validated inside the human tissue-mimicking gel and a chicken tissue sample. The average SAR values for the antenna were found in the safe range for using a multilayer metamaterial structure. The antenna showed better fractional bandwidth and gains in comparison to recent other implantable antennas. The antenna’s return loss was found entirely unchanged during the bending test. The SAR values for the flexible antennas are relatively higher than typical implantable antennas. So designing flexible implantable antennas is quite challenging. Multiband implantable applications in the MedRadio and ISM bands were covered in an earlier article [224] (Figure 13c). A complementary split-ring resonator (CSRR) was introduced in this antenna to decrease the antenna efficiency and gain. The antenna prototype was fabricated using a copper sheet, and the performance was measured using a pork phantom.

## 7. Flexible Antennas for Ingestible Application

Telecommunications and microelectronics have contributed a number of benefits in the field of medical applications. Ingestible medical devices (IMDs) are significant components for IoT applications in medical sectors. As a result, for monitoring devices and drug delivery system and monitoring internal condition of the patient, special types of flexible antennas are needed.

Wireless IMDs have been widely used for diagnostic purposes, in particular for visualization of gastrointestinal (GI) [225]. Because the digestive organs in the GI tract have different electrical properties, the antennas for these applications need to have broadband characteristics. In recent years, there are various antennas for wireless systems have been reported. However, heavy metals are mainly used to fabricate these antennas, which are potentially hazardous for humans’ health when the capsules fracture. Thus, water antennas appear safer in diagnosis and treatment. Wireless capsule endoscopy (WCE) is a technique for medical applications that records images of the digestive tract [226] (Figure 14). This method has various advantages, comparing to traditional methods, such as esophagogastroduodenoscopy or colonoscopy. WCE is painless and non-invasive. WCE system contains an antenna that offers wide bandwidth to scan different areas of the small intestine.

An omnidirectional conformal small UWB loop antenna operating at 433 MHz was proposed for capsule endoscope systems involving real-time video image transmission from inside to outside body [226]. The proposed antenna was fabricated on Preperm 255 substrate using copper, and it maintained the desired performance at varying implant depths and locations of different tissue types.

A compact (30 ${\mathrm{mm}}^{3}$), conformal differentially fed antenna on ultrathin polyimide substrate (Figure 15) at 915 MHz ISM (902–928 MHz) band for monitoring in body core temperature and a biomedical application was presented in an earlier report [227]. The integrated capsule system was experimented in a cubic homogeneous muscle phantom. Smaller capsule with differential network concept, the absence of shorting pins brought desirable advantages such as the insensitivity on antenna performance tuning, easy fabrication, and the reduction of effects on neighboring circuits. Recently, an electrically small wideband antenna on silicon substrate targeting wireless capsule endoscopy (WCE) application operating at 915 Hz was presented in an earlier report [228] (Figure 16). The antenna performance was satisfactory with different tissues due to its wide bandwidth, and it was also tested with liquid mixture mimicking the colon phantom at 915 MHz. This work paves the way for improved ingestible WCE by supporting higher data-rate radio links.

## 8. Performance of Different Types of Flexible Antennas

An antenna’s performance depends on various parameters like conductivity of the radiation element, dielectric substrates, and different design considerations. A highly conductive radiating element ensures superior gain, efficiency, and bandwidth of the antenna. Choosing a suitable dielectric material is critical for antenna performance. Efficiency and gain are reportedly reduced for a higher value of loss tangent of the dielectric substrate [229]. In addition, dielectric permittivity ($\varepsilon_{r})$ affects the bandwidth and the resonant frequency of the antenna. An increased permittivity value enables antenna miniaturization with reduced impedance bandwidth and low radiation losses [229]. Substrate thickness is another factor which can influence efficiency, gain, bandwidth, and directivity. For a flexible antenna, it is always a trade-off in choosing the proper substrate considering thickness, performance, and flexibility at the same time. Apart from the abovementioned factors, antenna patch design, array configurations, and power division transmission lines affect the antenna performance a lot. Patch elements come in various shapes like the rectangular, square, circular, annular ring, triangular, pentagonal, and square or circular with perturbed truncations [230]. These shapes affect polarization patterns, resonant frequencies, return loss, gain, and directivity. A computer-aided design (CAD) software is essential in combination with an electromagnetic wave solver to iteratively evaluate the design and the simulated antenna with different radiating material and substrate combinations.

Flexibility and bending due to mechanical stress for on-body measurement needs to be assessed to validate any flexible antenna performance in real world situation. Flexible antennas have to undergo mechanical deformation, such as bending or stretching. It degrades the antenna performance like shifting frequency, changing gain and radiation pattern, and changing antenna polarization for the intended application. For realizing IoT effectively, acceptable performance of flexible antennas are necessary. Research is still going on for trying to find creative ways to enhance antenna performance under stress or bending condition.

The effect of mechanical stress on a tunable and compact microstrip antenna on a polyimide substrate was investigated by performing bend and stretch tests [23]. The antenna was bent compressively with a minimum radius of curvature of 86 and 150 mm along the x-axis and y-axis, which resulted in a maximum increase of resonant frequency by 3.1% and 1.3%, respectively. Similarly, tensile bending was performed with a minimum radius of curvature of 79 and 162 mm along the x-axis and y-axis, which resulted in a maximum decrease of the resonant frequency by 4.2% and 0.3%, respectively (Figure 17). An overall 0.9% decrease in the resonant frequency was measured for an applied strain of 0.09% while stretching the antenna along the y-axis. Adhesion tests were also performed for checking the adhesion of the antenna with the substrate.

In an earlier report [55], the dipole antenna over the artificial magnetic conductor (AMC) structure maintained the reflection coefficient relatively the same regardless of the bending condition or the presence of a lossy host. The antenna’s radiation pattern remained broadside under bending condition and on a filled blood bag with 0.7 dBi gain variation. In another study [56], fabricated antenna for body area networks at 2.4 GHz had overcome the human body proximity detuning effect by varying radiator length. The multilayer inkjet-printed microstrip fractal patch antenna showed excellent stability and tolerance under different bending radii of curvature [60]. The inkjet-printed antenna reconfigurable antenna for WLAN/WiMAX wireless devices was fabricated and tested for both flat and curved geometries of different radii with well-maintained radiation characteristics [63]. The effect of antenna bending in x and y direction had been reported in [64] for reconfigurable dual-band dual polarized monopole antenna.

A numerical and experimental study of the impact of bending on the characteristic parameters of a flexible ultra-wideband (UWB) made of thin PEDOT polymer sheet on sticky tape substrate antenna was reported earlier [42]. Antenna characteristics, including S-parameters, polarization, and radiation patterns, were examined with regards to the bending angles from 0° to 180° (Figure 18). Though the antenna performance was found satisfactory even with acute bending angles, the polarization direction was changed.

The array configuration [81] exhibited a return loss of more than 15 dB in both frequency bands. Higher cross-polarization can be improved using different feeding networks at both layers. To demonstrate the flexibility and mechanical stability of the antenna arrays, they were flexed several times and recharacterized. The return loss and radiation patterns were unchanged within the repeatability of the measurement equipment.

The simulated and measured data of an mm-wave antenna and its antenna arrays performance regarding impedance bandwidth, radiation pattern, and realized gain for laser printed and inkjet-printed processes over Ka-band (26–40 GHz) was reported earlier [82]. The conformity analysis of the antenna was done by bending it along the cylindrical surface of different radii only for the reflection coefficient parameter (Figure 19).

Though inkjet-printed high-gain, multidirector Yagi-Uda antennas for use within the 24.5-GHz ISM band was fabricated on flexible LCP substrate, the antenna performance for the conformal application was not studied [95]. It is expected that antenna characteristics parameters change due to the interaction with the lossy human body tissues. Hence, the permittivity of human tissues and different conductivity values influence the reflection coefficients, affect the power absorbed by the body, and decrease the radiation efficiency of the antenna [231]. Moreover, the specific absorption rate (SAR) limit has to be considered while designing a wearable/flexible antenna. The SAR limit is regulated for wearable devices, including antennas, which quantifies the amount of EM radiation a human body can safely withstand without any health hazards and is defined as the power absorbed per unit mass of tissue [232].

The new miniaturized cavity-backed substrate integrated waveguide (SIW) textile antenna operating at 2.45 GHz was reported earlier [122]. In this work, the effects of bending the antenna were considered when calculating the reflection coefficient, gains of 5.28 and 5.35 dBi, and efficiencies of 73% and 74.3% in free space and on-body measurement which is appropriate for flexible wearable applications. SAR values for thigh, upper arm, and chest were found to be 0.297, 0.358, and 0.380 W/kg, respectively, which were less than 2 W/kg, which is the European limit. A low-cost wideband textile antenna based on the SIW technology on a woolen felt was presented in a previous article [120]. From the simulation, the SAR values of the antenna have been found below the American and Canadian limits of 1.6 W/kg on average. It was recommended to use this antenna for wearable applications like jackets without direct contact with the human body.

A 2.28–2.64 GHz wearable circular ring slot antenna had a maximum gain of 7.3 dBi in the ISM band, and the efficiency varied between 50% and 60%, and more than 70% with electromagnetic band gap (EBG) technique [233]. The antenna performance was investigated by placing it on different parts of the human body: arm, lap, and stomach. Figure 20 shows the experimental setup of the proposed antenna and $S_{11}$ parameter of the proposed antenna under these conditions. The SAR value with EBG structure was found to be 0.554 W/kg, met the SAR requirements for the US standards.

The dual-band textile patch antenna offered good coverage for WLAN operating bands fabricated using SIW technology on a felt substrate [234]. The antenna was bent with various bending radius to verify the performances. The total efficiencies of the array in the lower and upper bands were 55% and 60%, respectively. The average efficiencies were caused by the loss of the textile material. However, the upper band of the antenna maintained a good isolation as low as −35 dB, making it suitable for MIMO applications. The specific absorption ratio (SAR) value on the human body model, which was 0.067 W/kg, was far below the limit value 2 W/kg of European standard.

For the all-textile circular ring-slot antenna, a two-third muscle equivalent phantom was used in order to verify the performance on the human body [235]. The external effects were not able to influence the return loss characteristics of the antenna because of the SIW cavity-backed feed structure, including the clothing-mounted environment and human body. Figure 21 shows the simulation and measurement of the antenna for different bending radius setup. The radiation efficiency and measured peak gain were 37.7% and 3.12 dBi, respectively, which were reasonable for ISM applications. The radiation characteristic of the proposed antenna was insensitive to the phantom effect.

In another report [236], an ultra-wideband (UWB) antenna’s performance was investigated with changing of substrate’s thickness and dielectric constant, as well as bending along a cylindrical structure with 10 and 20 mm radii. It was found that there is not much difference in the bandwidth and the efficiency of the antenna under flat and bent cases which made it suitable in foldable Wireless Wide Area Network (WWAN) terminals, WBAN devices, and medical sensors. The convex and concave bending analysis was performed on an inkjet-printed multiband (covered GSM 900, GPS, UMTS, WLAN, ISM, Bluetooth, LTE 2300/2500, and WiMAX) antenna to characterize the flexibility [237]. For both types of configuration, the frequency shift was not that significant, a slight increase in the gain due to a slight rise in directivity was observed. No significant degradation was found, and the overall performance was satisfactory for various wireless applications for future conformal and flexible electronic devices.

Designing an antenna for a wideband operation makes it more immune to frequency shifting due to bending. If the antennas are designed for wideband operation, their resonant frequency could be kept within the required operating region even after bending [17]. Second, symmetrically shaped antennas are less affected by bending in different directions. Bending analysis is essential to test the durability or robustness of the antennas. Repeated bending tests were done to check physical deformations, discontinuities, or cracks on the conductive part of the antenna. Due to physical deformations or cracks on the radiating element, current density changes, which change the antenna polarization in effect. For wearable antennas, physical deformations increase the SAR value. Miniaturization of antennas can prevent physical distortions or crack due to bending. A performance comparison of the different flexible antennas for the last 3 years (2018–2020) has been shown in Table 3.

## 9. Challenges and Future Prospects of Flexible Antennas

Recently, the research on flexible wireless devices has attracted much attention because of its nature to comply with the requirements of biomedical applications, vehicular navigation systems, wearables, and so on. An antenna is one of the key components in this whole system, and for ensuring the device conformality, it should be flexible and stretchable. The first step towards this goal is to replace the conventional rigid substrates with flexible materials like textiles, paper, or elastomeric such as polydimethylsiloxane (PDMS) [241], PEN, PET, and PI. Thus, it can be said that the very first challenge of designing a flexible antenna is finding a suitable substrate. In comparison to the traditional substrates like FR4 or Rogers that have a dielectric constant around 3–10 and a loss tangent of 0.001–0.02, typical flexible substrates have low dielectric constants. Even though this low value of dielectric constant helps to achieve larger bandwidth and radiation efficiency, it creates a problem (antenna performance) when miniaturization is needed. For flexible textile antennas, the uneven thickness is another problem to deal with. The electro textile substrate is prone to crumble and susceptible to fluid absorption.

Paper-based flexible antenna faces similar problems with relatively high loss factor [90], which causes low antenna efficiency and impedance mismatch. In an earlier work [154], an organic paper-based UWB antenna was presented. Although it is a low-profile antenna, it is not the right choice for applications that require high levels of bending and twisting because of discontinuities and lack of robustness. A polymer-based substrate is an excellent option to solve these problems. For example, an earlier report [176] studied a compact polyimide-based antenna where Kapton polyimide film was used because of its low loss tangent (tan δ = 0.002) for broadband frequency operation with physical and chemical flexibility. This substrate had a temperature rating up to 400 °C and tensile strength of 165 MPa at 73 °F that confirms the robustness of the Kapton polyimide film. Furthermore, polyimide and Kapton are not very expensive because of roll-to-roll mass production while being transparent and bendable. There are many other polymer-based designs reported in the literature [238,240,242,243,244,245,246]. One problem that can arise from the polymer-based antenna is excessive bending or twisting that might result in micro-cracks in the substrate. This will affect the electrical conductivity of the antenna and raise the risk of breakdown. Further, low glass transition temperatures of polymer make them unusable in high-temperature applications. Ceramic substrates can be an alternative which can withstand high temperature and can be used in flexible applications [247].

Such a limitation can be overcome by embedding very thin metallic nanowires on the surface of elastomers like PDMS to make it highly conductive and stretchable [37]. Because of the fabrication and design complexity, it is not very suitable for low-cost, flexible applications. Instead of solid metal wire, if liquid metal (LM) is used in the microfluidic channel created by elastomers, it will give the antenna reconfigurability, an exciting feature of antenna that is needed in many applications. PDMS is the most popular commercial elastomers to make the microfluidic channel for the flexible antenna. Different liquid metals such as mercury, carbon nanotubes (CNT), Galinstan, gallium indium (GaIn), and eutectic gallium indium (EGaIn) are injected into the channel to form the antenna [248,249,250]. Besides PDMS, EcoFlex silicone rubber [251] and thermoplastic polyurethane (TPU) based NinjaFlex [114] are also used as an elastomer for creating microfluidic channel, and are usually 3-D printed to realize a specific pattern. Another challenge of designing flexible antennas is identifying suitable conducting materials that sustain different bending and twisting conditions and have reasonable resistance value as not to affect the antenna radiation efficiency. Various methods have been considered to find the conductive substrates, such as chemically modifying fabrics surface [252] or physical mixing of several conductive materials [253,254,255].

Future flexible antennas should feature low profile, low loss, easily integrable with RF front end system, ability to control/manipulate radiation pattern, and eventually circular polarization for wider bandwidth. Regardless of the shape of the flexible antenna, one way to downsize the antenna structure is to exploit the higher frequencies of V (40–75 GHz) and W bands (75–110 GHz) [256]. This high-frequency operation will ensure a high-performance data connection. Materials with high dielectric constants are used to miniaturize the size of the antenna [257,258]. Most common elastomeric materials have low dielectric constants. This low value can be increased by mixing the substrate with high dielectric constant materials such as ceramics like Ba_x_Sr_1−x_TiO_3_ [258]_,_ BaTiO_3_ [257], NdTiO_3_ [259], MgCaTiO_2_ [259], CNTs [260], and nanoparticles [261]. Metamaterial based flexible antenna is a relatively new development and has found its way in the commercial market because of its characteristics like lightweight, robustness, and reconfigurability [239,262,263,264,265]. It has the potential to be cheaper and smaller. The co-design of antenna and RF system on a flexible substrate has made breakthroughs in biomedical implantable devices [122]. Improvement in the design techniques and introduction of new materials will help make this co-design system more viable for many other applications. Metamaterial has a natural ability to couple with the radiation and converts it from one type of energy to another. This feature of the metamaterial can be used to flexible rectenna for energy harvesting [266,267].

## 10. Flexible Antenna for Future Wireless Solutions

Flexible antennas for future wireless solutions are expected to work in a broad range of frequencies due to the increased demand for wireless applications such as the Internet of Things (IoT), body area network (BAN), and biomedical devices. There are different antenna methodologies, single-band antenna, multiband antenna, and reconfigurable antenna. Multiband design is often necessary, for example, devices in wireless LAN should operate in both 2.4 and 5 GHz range. In addition, the design should ensure that the antenna’s characteristics stay consistent under bending conditions. A low-cost inkjet-printed multiband antenna was developed in an earlier article [236]. A novel triangular iterative design with coplanar waveguide (CPW) feed printed on Kapton polyimide-based flexible substrate was used to achieve multiband operation with wide bandwidth. The antenna covers the GSM 900, GPS, UMTS, WLAN, ISM, Bluetooth, LTE 2300/2500, and WiMAX standards. Concave and convex bending was used to evaluate the antenna. Convex bending shows no significant resonance frequency shift, while during concave bending, there is a maximum 3% shift. A planar inverted-F antenna (PIFA) made of a flexible printed circuit (FPC) with multi-band operation available for Bluetooth and IEEE 802.11a/b/g standards were developed earlier [268].

The antenna’s characteristics stay consistent while the angle of folding is less than 90 degrees. Flexible and wearable antennas were designed in [269] for wireless and satellite-based Internet of Things (IoT) and wireless body area network (WBAN) applications. The antenna operates in the C-band (4–8 GHz) for satellite communication to avoid congestion in lower frequency satellite bands.

There are different types of reconfigurable antennas, including polarization, frequency, and reconfigurable pattern antenna. The significant benefit of a reconfigurable antenna is its capability to switch bands based on the end-user’s application requirements. In a previous work [270], a flexible, spiral-shaped frequency reconfigurable antenna is developed that covers aeronautical radio navigation, fixed satellite communication, WLAN, and WiMAX standards. Frequency reconfiguration is achieved by the incorporation of a lumped element in the strip so that the antenna can switch between different resonances. A flexible, reconfigurable antenna using polarization was proposed in earlier work [271] (Figure 22b). The intended use for the antenna is in biomedical applications as a remote patient monitoring system operating in WBAN and WiMAX standards. In another article [272], a wearable pattern-reconfigurable antenna was proposed. The inductor loaded patch antenna can change its resonance between zero-order resonance and the +1 resonance, yielding two different radiation patterns. The antenna was designed to operate in the 2.4 GHz band.

Ultrawideband (UWB) technology allows for efficient bandwidth utilization using spectrum overlay (often referred to as shared unlicensed) with transmission power control. By restricting the transmission power, devices can operate in 3.1–10.6 GHz without causing interference. Therefore, UWB technology is attractive for wireless indoor and wearable applications. In a prior article [273], a wearable band notched UWB antenna is proposed. The band notch is included to avoid interference from WLAN applications, as is recommended for wearable and indoor UWB applications in the IEEE 802.11a standard. Antenna properties have negligible variations when bent at different angles and can withstand extreme conditions. A flexible and transparent UWB antenna was proposed in another study [49]. The antenna consisted of a transparent conductive tissue integrated with polydimethylsiloxane (PDMS). The antenna operates between 2.2 and 25 GHz. No significant performance degradation is measured under folding.

In addition to antennas being small, flexible, and capable of operating in a broad verity of wireless standards, some applications require devices to be battery and wire-free. Energy scavenging using a rectifying antenna (rectenna) can be leveraged to develop autonomous devices. A rectenna works by foraging RF energy emitted by radio transmitters. In another study [177], wireless power transfer was used in an RF-powered leadless pacemaker. The authors proposed a novel wideband numerical model (WBNM). Tissue simulating liquid (TLS) was used to design the wideband numerical model. The model was validated experimentally and analytically using a microstrip patch antenna. Furthermore, a novel metamaterial-based conformal implantable antenna operating in the frequency of 2.5 GHz was developed. The authors validated the antenna as a potential candidate for future RF harvesting leadless pacing applications.

A dual band printed planar antenna was proposed for microwave power transfer (MPT) in an earlier study [274]. The antenna was suitable for wearable applications and operated at ultra-high frequencies of 2.5 and 4.5 GHz. The antenna’s area is 15 mm × 14 mm × 0.17 mm, and Kapton polyimide-based flexible substrate and FR-4 substrate was used for the receiving antenna and the transmitting antenna, respectively. Safety is considered where the antenna meets the specific absorption rate (SAR) requirements. A planar inverted-F antenna (PIFA) and a rectifier circuit for far-field wireless power transfer were proposed in [275]. The compact implantable rectenna can operate at a frequency of 2.45 GHz. To increase the power level, a parasitic patch attached to the human body was used to enhance the directivity of the rectenna. Safety is considered where the Federal Communications Commission (FCC) limits for radiating antennas, Specific Absorption Rate (SAR), and temperature increase were tested to assure compliance of the rectenna with international safety regulations. In a prior report [276], a hybrid energy harvesting circuit combining a solar cell and electromagnetic energy (EM) harvesting rectenna is proposed (Figure 22a). A flexible polyethylene terephthalate (PET) substrate together with a flexible amorphous silicon solar cell was used to achieve both low cost and conformal structures. The antenna measures 13 mm × 6.5 mm. Both wideband and multiband topologies are presented. To achieve energy harvesting using a variety of communication standards, the antenna is capable of operating in frequencies between 800 and 6 GHz.

Millimeter-wave communication systems are anticipated to resolve the problems of congestion, lower bandwidth, and high latency in the current wireless systems. The upcoming 5G technology is expected to address these problems and offer higher channel capacity with wider bandwidth that ensures a higher data rate. Many smart devices are expected to have a high-speed uninterrupted internet connection and have irregular shapes. For this type of application, flexible and stretchable antennas are required that can be mounted on a conformal structure. A PET-based flexible T-shaped mm-wave antenna is proposed in [12] that is covering frequency from 26 to 40 GHz. Defected ground plane (DGS) is used to extend the bandwidth of the antenna by combining multiple resonant points. Impedance matching is not good at lower frequency range when the heat sintering has been done, and the performance of the antenna in bending condition has not been experimentally validated. In [277], the authors designed a transparent and flexible PET-based Y-shaped antenna that operates from 23 to 29.5 GHz depending on the angle of the arm of Y shape. For ensuring the conformity of the antenna, AgHT transparent conductive material is used in this research. A parametric study has also been conducted on how changing the angle of the arms of the Y shaped antenna affects the resonant frequency and the bandwidth. The study shows that changing the Y shaped structure to T shaped structure gradually lowers the resonant frequency from 27.71 to 24.60 GHz. It has been claimed as a reconfigurable antenna, that is not justified in the article it how it can be tuned dynamically.

In another study [278], a dual-band wearable antenna for smartwatch is proposed based on new flexible material “ULTRALAM^®^3850HT”, and operates at 38 and 60 GHz. It is a rectangular antenna with six U shaped slots in the radiating patch. The response of the antenna is verified using two commercial software HFSS and CST. No experimental validation has been provided in this research. 5G technology is aiming at tackling the vastly growing and developing IoT domain. Many novel and future solutions, such as integrated wearable devices, household appliances, industry solutions, robotics, self-driving cars, and other solutions, are expected to benefit from the 5G network. Another equally important aspect of the forthcoming 5G is the ability to manage vast amounts of ‘always-connected’ IoT devices. Flexible antennas are a key component for realizing future wireless solution leveraging the 5G technology.

## 11. Conclusions

The field of flexible antennas is fascinating and interdisciplinary involving electrical engineering, materials science, and mechanical engineering. Flexible antennas are one of the critical components in the realization of flexible electronic devices. The flexible antenna is ideal for current and futuristic wireless communication and sensing applications primarily due to its lightweight, reduced form factor, low-cost fabrication, and ability to fit non-planar surfaces. The choice of materials for antenna fabrication is based on application preferences such as environment, seamless integration with rigid and non-rigid devices, cost, and mass manufacturing aspects of the fabrication process. Highly conductive materials such as Ag nanoparticles inks, Cu tape or clad, conductive polymers, PDMS embedded conductive fiber, and graphene-based materials have been typically used to implement the conductive patterns in the antenna.

Kapton polyimide, PET, PEN, PANI, liquid crystal polymer, electro-textile, and paper have been preferred as flexible substrates. Applications of flexible antennas in the different frequency band below and above 12 GHz indicates the versatility of flexible antennas. Different miniaturization techniques have been discussed, along with challenges and limitations. Flexible antennas for biomedical applications with implantable and ingestible functionality shows the promising nature of the electromagnetic devices in health care. Bending, stretching, and proximity to the human body’s impact on flexible antennas performance has been discussed. Corrective measures such as increasing the bandwidth or symmetry of the designed antenna can help account for deviations caused by deformation and other spurious factors.

Finally, challenges for designing and realizing a flexible antenna has been discussed considering material challenges for the substrate and the conducting material. Flexible antennas for the future wireless system as a part of IoT, BAN, and biomedical devices have been reviewed with the citation of recent literature. The latest research on flexible antennas with emphasis on power sustainability via energy harvesting is discussed. Despite the limitations of flexible antennas, these non-rigid devices can be engineered to meet the futuristic demand for a compact wireless solution to fit the surface of any curvature.

## Figures and Tables

**Figure 1 micromachines-11-00847-f001:**
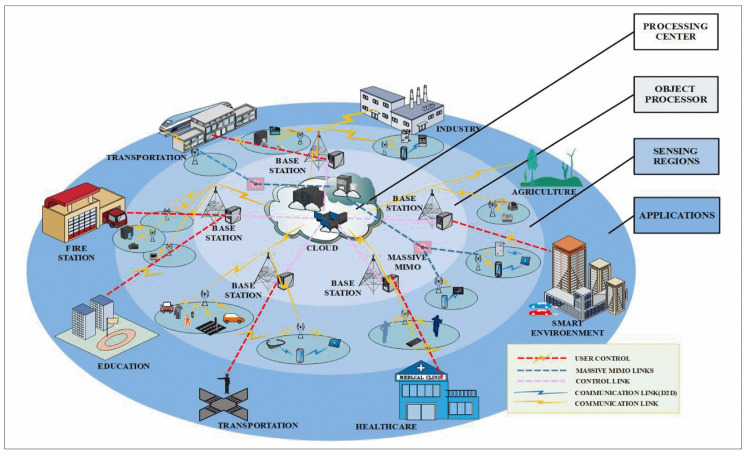
Connection architecture between 5G and Internet of Things (IoT). Reprinted with permission from Ref [5].

**Figure 2 micromachines-11-00847-f002:**
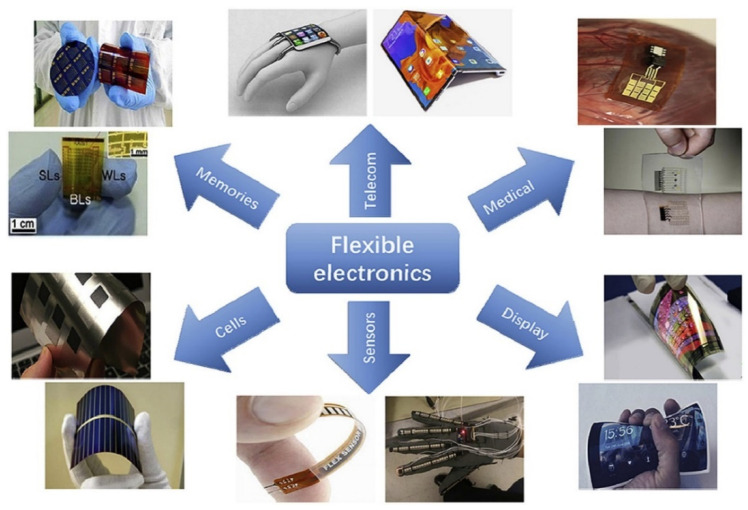
Application areas for flexible electronics. Reprinted with permission from Ref [9].

**Figure 3 micromachines-11-00847-f003:**
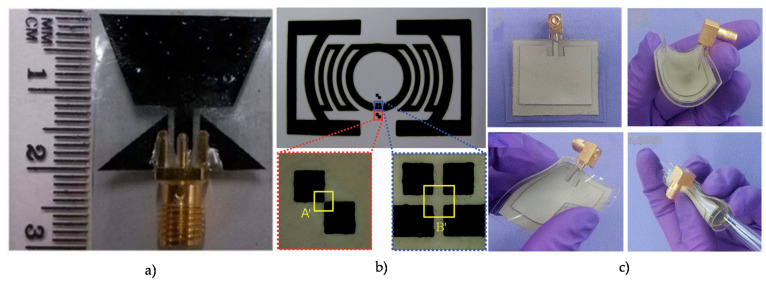
Schematic of antennas with different conducting materials. (**a**) Polymer ultra-wideband antenna using poly(3,4-ethylenedioxythiophene) polystyrene sulfonate (PEDOT:PSS) [42], (**b**) platinum-decorated carbon nanoparticle/polyaniline hybrid paste for flexible wideband dipole tag-antenna [30], and (**c**) a stretchable microstrip patch antenna composed of nanowire (AgNW)/polydimethylsiloxane (PDMS) flexible conductor [37].

**Figure 4 micromachines-11-00847-f004:**
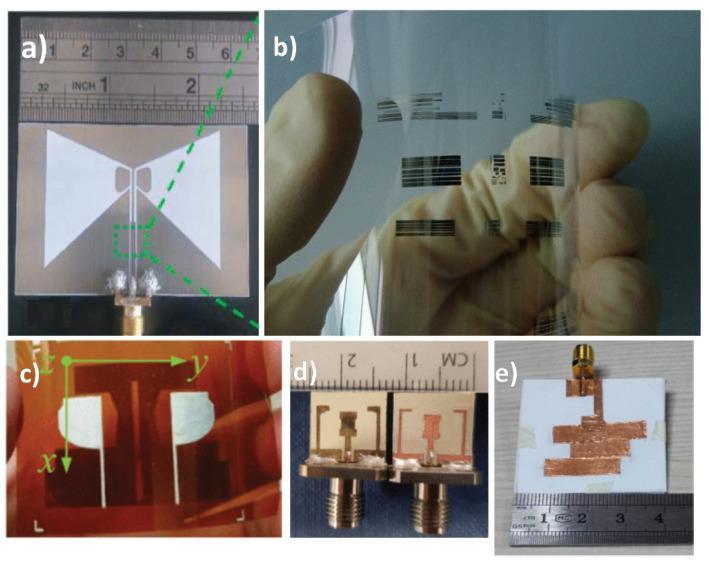
Fabricated antenna prototypes on: (**a**) polyethylene terephthalate (PET) [67], (**b**) polyethylene naphthalate (PEN) [70], (**c**) Polyimide [59], (**d**) liquid crystal polymer (LCP) [82], and (**e**) paper [80] substrates.

**Figure 5 micromachines-11-00847-f005:**
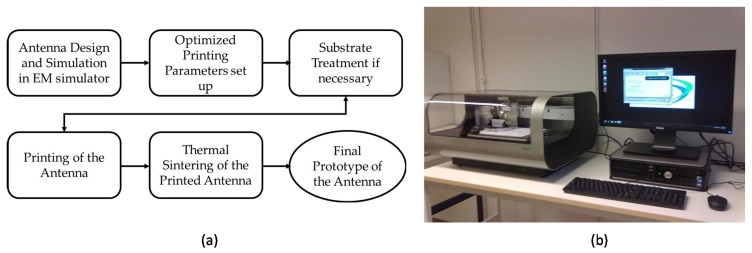
Overview of the inkjet printing process for antenna fabrication. (**a**) Flowchart of the inkjet printing process [91] and (**b**) Dimatix Materials Printer, DMP-2800 (FUJIFILM Dimatix Inc., Santa Clara, CA, USA), and the PC that is used to control the printer.

**Figure 6 micromachines-11-00847-f006:**
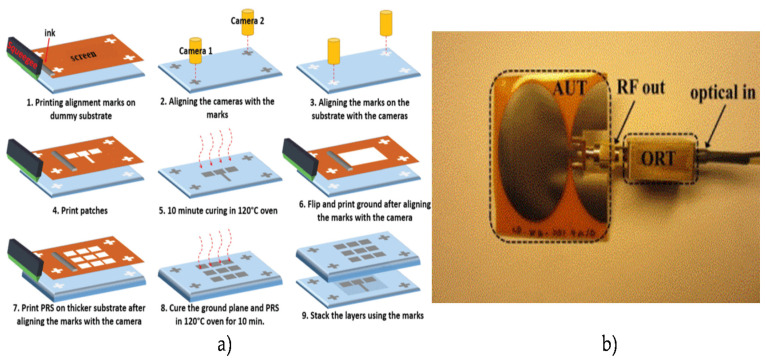
Schematic of the screen-printed antenna. (**a**) Fabrication process for parasitic beam-switching millimeter-wave antenna array [104] and (**b**) screen printed graphene flakes based wideband elliptical dipole antenna prototype [105].

**Figure 7 micromachines-11-00847-f007:**
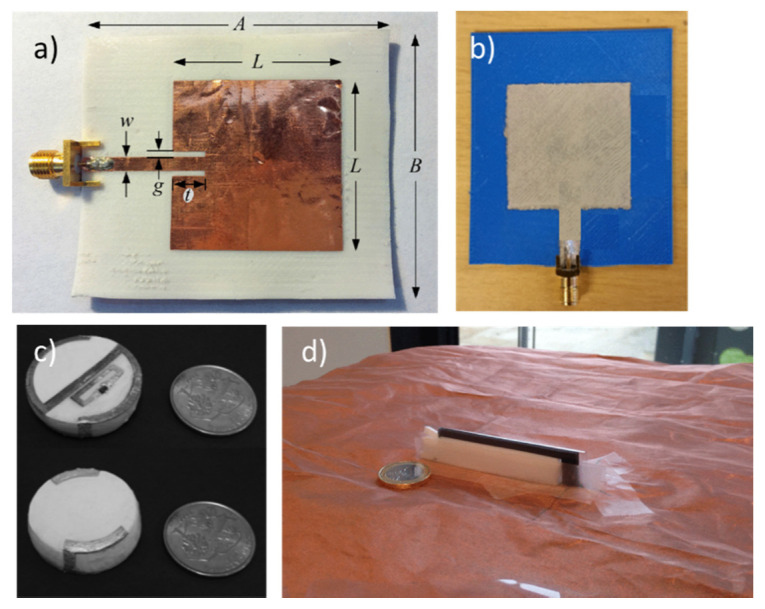
Examples of D printed antennas. (**a**) Square patch antenna on NinjaFlex substrate (dimensions in mm A = 65, B = 55, L = 35.8, w = 3, g = 1, t = 7) [107], (**b**) brush-painted wearable antenna on a 3-D printed substrate [112], (**c**) button-shaped 3-D radio-frequency identification (RFID) tag antenna [111], and (**d**) 3-D printed flexible inverted-F antenna (IFA) [113].

**Figure 8 micromachines-11-00847-f008:**
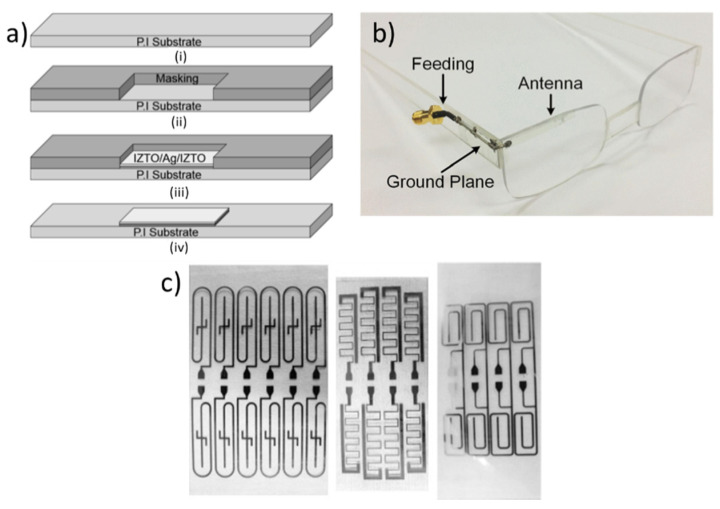
Example of wet etching based antenna structure. (**a**) Transparent and flexible antenna fabrication process: (i) polyimide (PI) substrate cleaning; (ii) masking. (iii) deposition; and (iv) mask removal [116] and (**b**) fabricated indium–zinc–tin oxide (IZTO)/Ag/IZTO (IAI) antenna [116] and (**b**) Cu thin film RFID ultra-high frequency (UHF) antenna on PET using photolithography and sputtering [118].

**Figure 9 micromachines-11-00847-f009:**
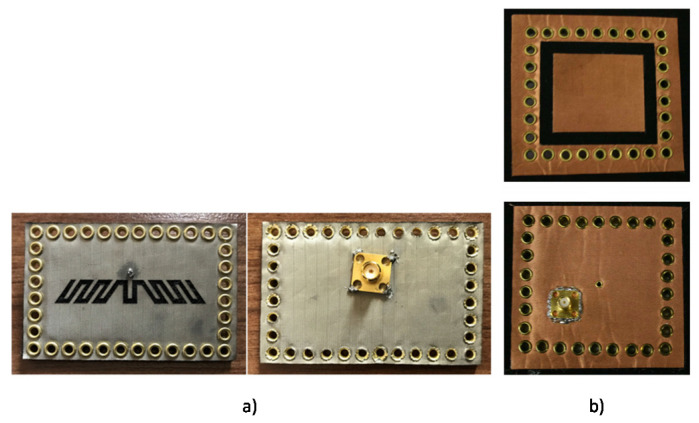
Antennas fabricated using the substrate integrated waveguide (SIW) method. (**a**) The prototype of the SIW antenna using conductive fabrics [122] and (**b**) prototype of the circularly polarized SIW antenna [121].

**Figure 10 micromachines-11-00847-f010:**
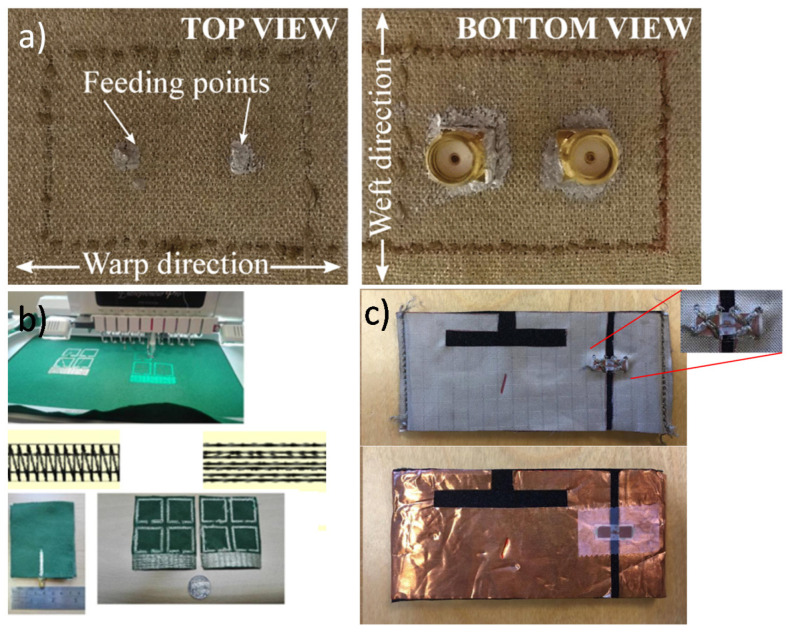
Embroidery and stitching based flexible antennas. (**a**) Novel mixed embroidered-woven textile integrated waveguide (TIW) antenna: top and bottom views [128], (**b**) embroidery metamaterial antenna manufacturing process with stitch pattern embroidery layout and embroidered antennas [129], and (**c**) fabricated e-textile based on a slotted patch antenna created with a sewing machine and copper tape [127].

**Figure 11 micromachines-11-00847-f011:**
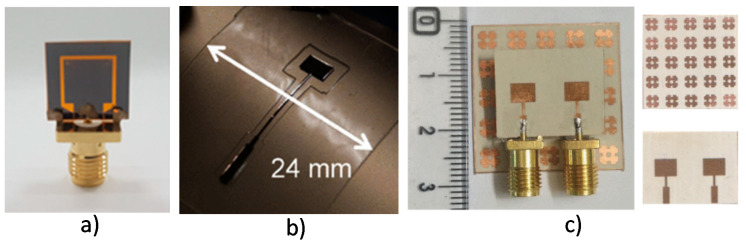
Flexible antennas for applications above 12 GHz. (**a**) Graphene antenna prototype for 5G applications [179], (**b**) millimeter-wave planar antenna prototype on PDMS substrate [185], and (**c**) electromagnetic bandgap (EBG) backed millimeter-wave multiple-input-multiple-output (MIMO) antenna for wearable use [186].

**Figure 12 micromachines-11-00847-f012:**
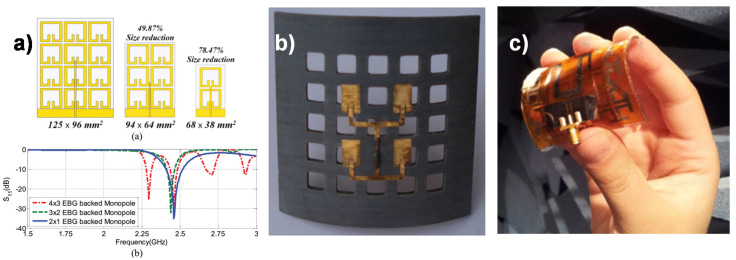
Miniaturized antennas. (**a**) Miniaturization of EBG-backed monopole antenna with reflection coefficient [212], (**b**) fabricated cylindrical conformal array antenna with photonic bandgap (PBG) lattice [221], and (**c**) Prototype of an M-shaped printed monopole antenna and slotted Jerusalem Cross (JC)-artificial magnetic conductor (AMC) [208].

**Figure 13 micromachines-11-00847-f013:**
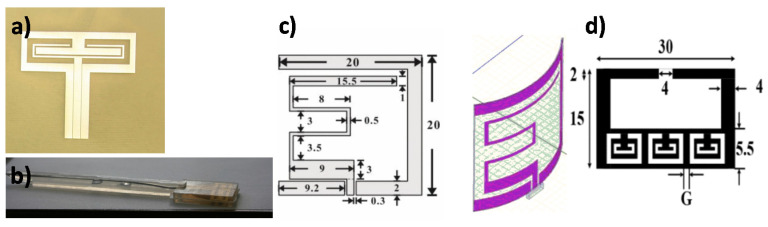
Implantable antennas. (**a**) Implantable slot antenna, (**b**) embedded in PDMS [222], (**c**) Geometry of low specific absorption rate (SAR) antenna and its prototype [223], and (**d**) geometry of flexible implantable loop antenna with complementary split ring resonator (CSRR) [224].

**Figure 14 micromachines-11-00847-f014:**
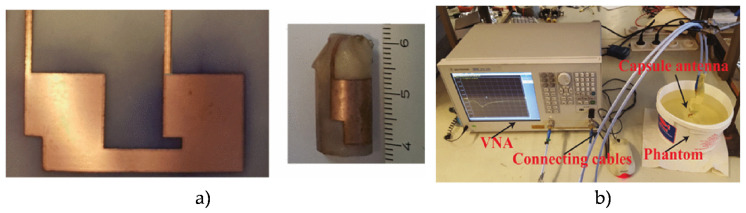
Antenna for ingestible application. (**a**) Fabricated capsule antenna and (**b**) measurement setup using Agilent e5063A vector network analyzer [226].

**Figure 15 micromachines-11-00847-f015:**
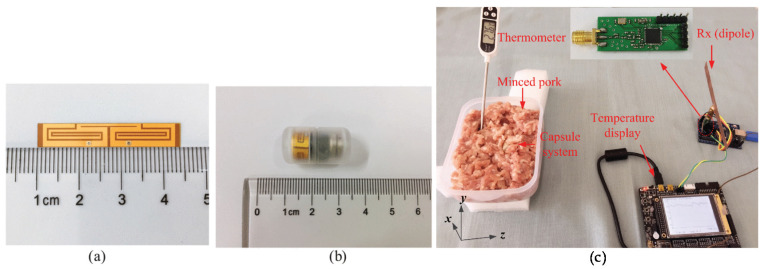
Prototypes of antenna for biomedical applications. (**a**) Proposed antenna, (**b**) integrated capsule system, and (**c**) measurement setup of communication system [227].

**Figure 16 micromachines-11-00847-f016:**
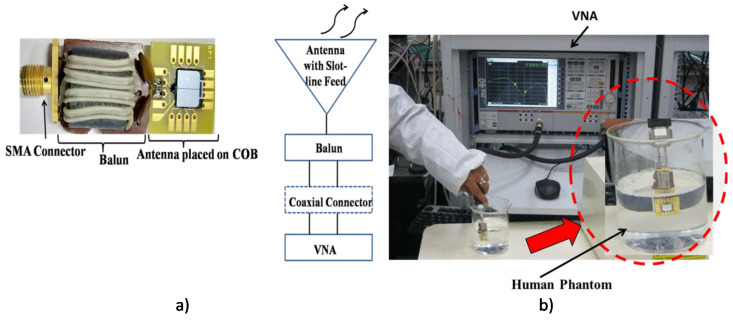
Antenna for wireless capsule endoscopy (WCE) application. (**a**) Antenna Assembly and (**b**) measurement set up for fabricated antenna [228].

**Figure 17 micromachines-11-00847-f017:**
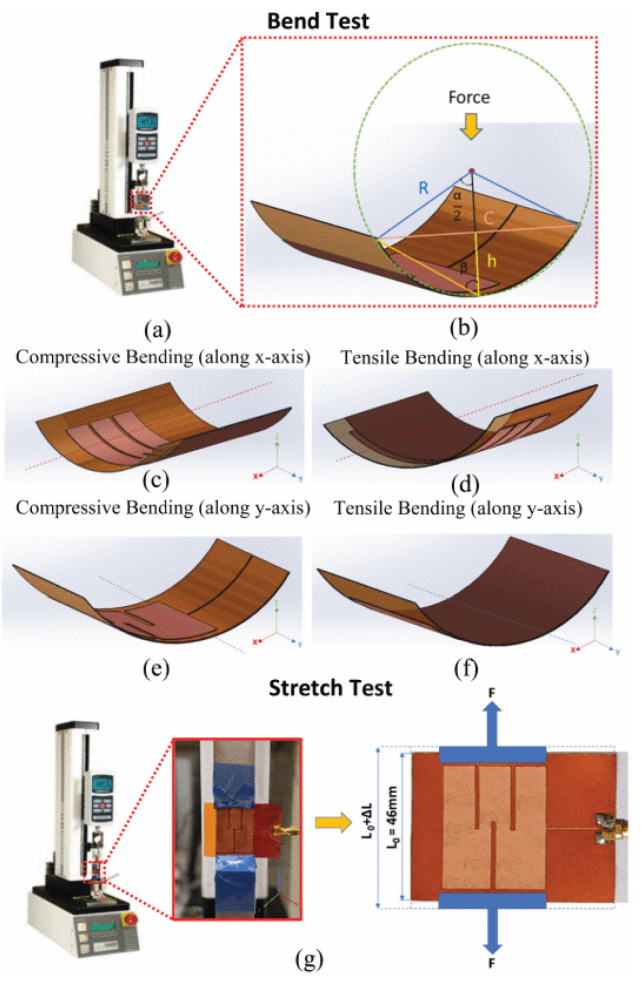
Experiment process of compressive and tensile bending and stretching the antenna along the x-axis and y-axis [23].

**Figure 18 micromachines-11-00847-f018:**
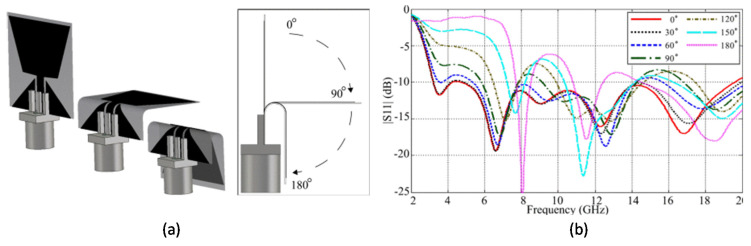
Impact of bending on antenna performance. (**a**) 3-D bending setup and definition of the bending angle in CST microwave studio, and (**b**) simulated |*S*_11_| for different bending angles [42].

**Figure 19 micromachines-11-00847-f019:**
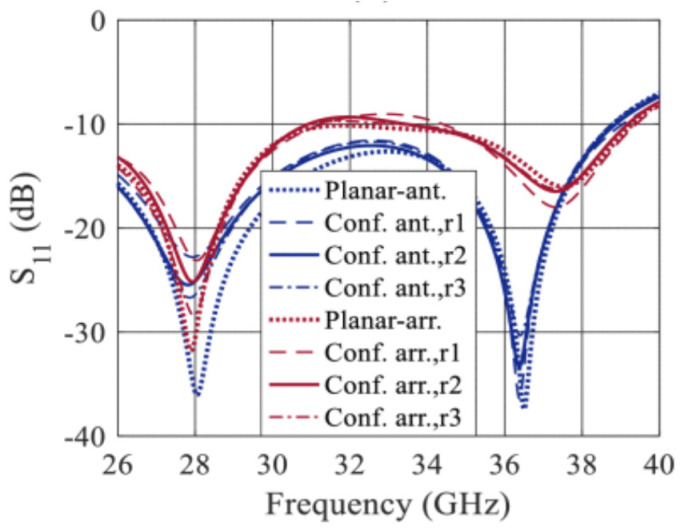
Comparative return loss of planar vs. conformal antenna configuration along the cylindrical surface of radii, *r*_1_ = 6 mm, *r*_2_ = 8 mm, and *r*_3_ = 10 mm [82].

**Figure 20 micromachines-11-00847-f020:**
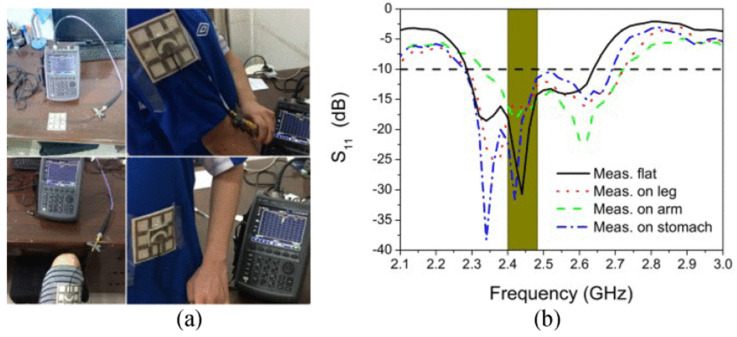
Measured $S_{11}$ of the proposed antenna placed on human tissue. (**a**) Measurement setup and (**b**) $S_{11}$ curves [233].

**Figure 21 micromachines-11-00847-f021:**
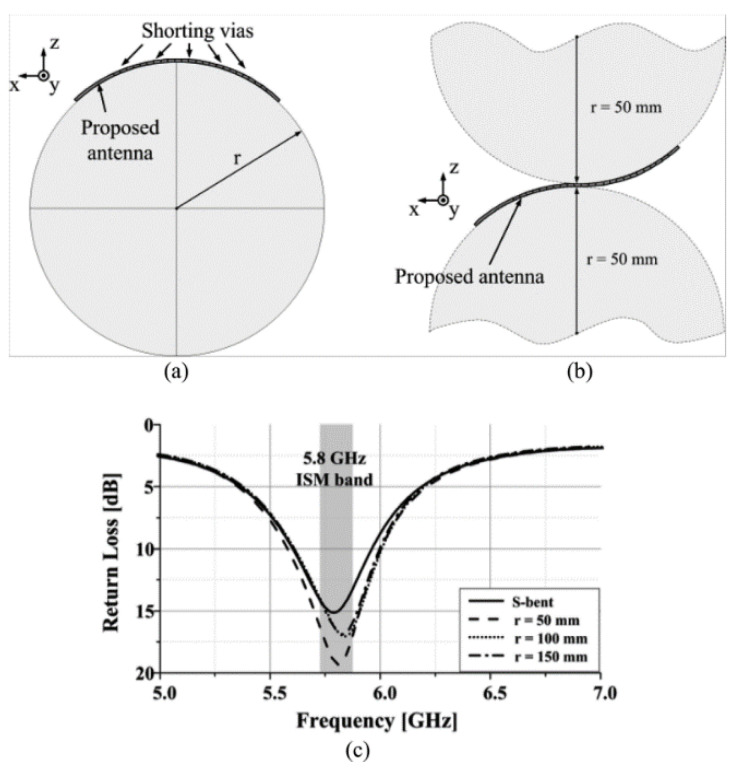
Antenna bending simulation. (**a**) Bending setup. (**b**) S-bending setup. (**c**) Return loss characteristics [235].

**Figure 22 micromachines-11-00847-f022:**
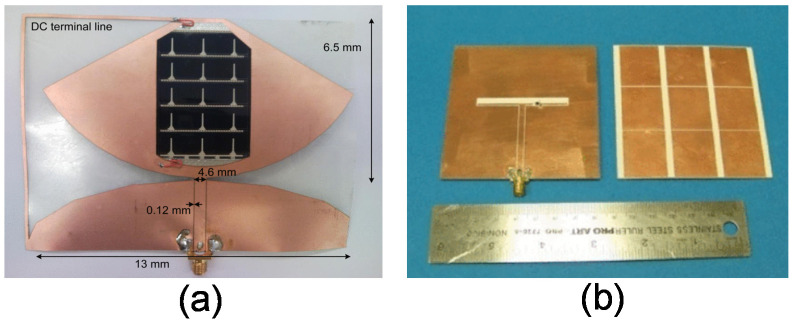
(**a**) Hybrid solar and electromagnetic energy (EM) energy harvesting antenna on PET substrate [263] and (**b**) reconfigurable antenna with AMC surface [271].

**Table 1 micromachines-11-00847-t001:** Conductive materials and their electrical conductivity values.

Materials Types	Conductive Materials	Conductivity, *σ* (S/m)
Metal nanoparticles	Ag nanoparticle [43]	2.173 × 10^7^
	Cu nanoparticle [44]	1 × 10^6^
Conductive Polymers	PEDOT:PSS [45]	100–1500
	Polyaniline (Pani) [45]	5
	Polypyrrole (PPy) [45]	40–200
Conductive Polymers with additives	C nanotube [46]	4000–7000
	PANI/CCo Composite [47]	7.3 × 10^3^
	AgNW/PDMS [23]	8130
	Ag flakes + Fluorine Rubber [24]	8.5 × 10^4^
Graphene Based materials	Nanoflakes [48]	6 × 10^5^
	Paper [31]	4.2 × 10^5^
	Meshed Fabric [49]	2 × 10^5^
Liquid Metal	Eutectic GaIn [40]	3.4 × 10^6^

**Table 2 micromachines-11-00847-t002:** Commonly used flexible substrates with their dielectric constant, loss tangent, and thickness.

Substrates		Dielectric Constant $\left(\varepsilon_{r}\right)$	Dielectric Loss (tanδ)	Thickness (mm)
PET [63]		3	0.008	0.140
PEN [70]		2.9	0.025	0.125
Polyimide [61]		2.91	0.005	0.2
PDMS-MCT [84]		3.8	0.015	-
PDMS [73]		2.65	0.02	-
PDMS with glass microsphere [73]		1.85	0.014	-
PDMS with phenolic microsphere [73]		2.24	0.022	-
PDMS with silicate microsphere [73]		2.45	0.02	-
Paper (Kodak Photo paper) [79]		2.85	0.05	0.254
Liquid Crystal Polymer (ULTRALIM 3850) [62]		2.9	0.0025	0.1
Wearable antenna substrates	Fleece Fabric [85]	1.25	-	2.56
	Cordura [86]	1.1–1.7	0.0098	0.5
	Woolen felt [87]	1.16	0.02	3.5
	Felt [88]	1.3	0.02	1.1
	Cotton/Polyester [89]	1.6	0.02	2.808

**Table 3 micromachines-11-00847-t003:** Flexible antenna performance comparison of recent investigations.

Reference No:	Antenna Type:	Dimensions mm^3^and Application	Substrate, Conductive Element	Bandwidth(FBW%)	Resonant Frequency	Bandwidth and Resonant Frequency under Bending	Antenna Gain and Efficiency
Normal Condition	Under Bending
[23]	Microstrip Patch	65 × 46 × 0.127,ISM band application	Kapton Polyimide, Flexible Copper Tape		900 MHZ	Increased 3.1% and 1.3% (along x-axis and y-axis) for compressive bending and decreased 4.2% and 0.3% (along x-axis and y-axis) for tensile bending	N/A	N/A
[24]	Microstrip-based Koch fractal	39 × 39 × 0.508, WBAN applications	Vinyl polymer based flexible substrate, Cu	2.36–2.55 GHz	2.45 GHz.	A slight shift in resonant frequency, bandwidth remained almost identical.	2.06 dBi. 75%	−0.57 dBi (on-body)
[28]	Microstrip patch.	60 × 60 × 0.110, C-band and future organic electronics applications	Rogers RT/Duroid ^®^ 5870, PANI/MWCNTs	4.43–4.76 GHz (7.33% width)	4.5 GHz.	N/A	5.18 dB	N/A
[33]	Elliptical quasi-dipole antenna	46 × 45 ${\mathrm{mm}}^{2}$, 2 up to 5 GHz for low-cost wireless communications applications.	Kapton Polyimide, graphene flakes.	1–5 GHz	2 GHz.	32,500 bending cycles decrease the resistance of the graphene flakes—no mention of performance analysis of the antenna.	2.3 dBi at 4.8 GHz, 56 ± 5%.	N/A
[60]	Multilayer microstrip fractal patch antenna	22 × 31 × 0.125, On-package, and on-chip printed antennas.	Kapton Polyimide, Ag NP	4.79–5.04 GHz.	N/A	Bandwidth for different bending radius was a bit wider.	Peak gain 4.5 dBi	Gain increase of 0.3 and 0.4 dB at 4.9 GHz for bending.
[67]	CPW-fed Bowtie Slot antenna	64 × 42 × 0.135, WLAN, WiMax, 3G, and 4G application.	PET, Ag NP	2.1 GHz to 4.35 GHz (69.77%)	2.1, 3.3, and 4.1 GHz	Bandwidth shifted to the lower frequency region	6.3 dBi at 4.35 GHz	N/A
[68]	CPW-fed Slotted Disc monopole	40 × 38 × 0.135, ISM band, suitable for early detection of brain stroke	PET, Ag NP	2.25–2.73 GHz (19.55%)	2.45 GHz	Return loss decreased −5 dB at 2.45 GHz. The resonant frequency did not shift for different bending radius.	2.78 dBi at 2.45 GHz.	2.51 dBi at 2.45 GHz
[80]	Microstrip patch antenna	40 × 35 × 0.6, Intrabody telemedicine systems in the 2.4 GHz ISM bands	Photo Paper, Cu strips	2.33–2.53 GHz (8.33%)	2.43 GHz.	Reflection coefficient remained adequately consistent with various bending radii, and a slight change in frequency occurred	2 dBi, 83% efficiency	More than 2 dBi, 70% efficiency on human phantom
[82]	CPW-fed rectangular patch with tapered sides and slots	11 × 12 × 0.1, Future flexible 5G front ends and mm-wave wearable devices.	Rogers ULTRALAM 3850 LCP substrate, Ag NP.	26–40 GHz	28 GHz, 38 GHz.	Bandwidth and resonance frequency remains conserved even if the prototypes are folded or bent for conformal integrations.	Peak gain 11.35 dBi at 35 GHz and above 9dBi for entire Ka-band	N/A
[111]	Dipole RFID tag antenna	Radius 15 mm, height 7.5 mm, RFID applications for wearable devices in FCC band.	Button shaped ABS substrate, Ag NP	910–925 MHz	920 MHz	N/A	Peak gain of −5.6 dB and maximum reading range of 2.1 m with a total transmitted power of 4.0 W	N/A
[161]	Monopole antenna etched with three elliptic single complementary split-ring resonators.	27 × 21 × 0.068, Ultra-wideband band-notched antenna.	LCP substrate, laminated Cu	3.7–4.2 GHz, 5.15–5.35 GHz, and 5.725–5.825 GHz(Notched frequency band)	N/A	The center-frequencies shift a little to a higher frequency for all the notched bands	2.41 and −0.44 dBi at 4.6 and 6.2 GHz	N/A
[163]	Slotted Monopole Patch	24 × 28 × 1.524	PDMS composite, conductive fabric	3.43–11.1 GHz (59.9%)	N/A	Bandwidth varied a maximum of 1% variation	2–4 dBi	N/A
[165]	CPW-fed H-shaped slot antenna	32 × 52 × 0.28, UWB antenna for wearable applications.	Flexible ceramic substrate ( = 3.2), graphene assembled film (GAF)	4.1–8.0 GHz (67%)	4.45, 5.6, and 7.1 GHz	Both the lower bandwidth and the upper bandwidth broaden as the antenna bends, whereas the radiation efficiency is not noticeably affected. Resonant frequency heavily shifted for on-body applications.	3.9 dBi at 7.45 GHz	4.1 dBi at 7.45 GHz
[179]	CPW-fed rectangular slot with chamfer	11.8 × 12.2 × 0.1, Mobile terminal for Fifth Generation (5G)	Kapton polyimide, Graphene ink	14.30–15.71 GHz (9.40%)	13.8 GHz	N/A	9..28dBi, 67.44%	N/A
[189]	Multi-slot antenna with full ground plane.	85 × 60 × 4.1, Wearable EM head imaging system.	PDMS-*Al_2_O_3_*-G composite substrate, Cu foil	1–4.3 GHz (124%)	1.13 GHz, 3 GHz.	Bending changed resonant frequencies but bandwidth remained almost identical	N/A	N/A
[233]	Circular ring slot antenna with EBG structure	81 × 81 × 4, wearable applications in ISM band	Wool felt, conductive textile “Nora-Dell-CR Fabric.”	2.28–2.64 GHz	2.45 GHz	*S*_11_ curves differed slightly, but the bandwidths covered WBAN band for placing leg, arm, and stomach of the human body.	7.3 dBi peak gain in the ISM band, 70% efficiency when EBG added.	N/A
[235]	CPW-fed Hybrid Shaped patch	30.4 × 38 × 70, WWAN terminals, WBAN devices, and medical sensors.	Kapton polyimide, Copper	3.06–13.58 GHz, 15.9–20.5 GHz, and 20.9–22 GHz	3.5, 6.7, and 12 GHz	Bandwidth for 20 mm curvature is 2.8 to 13.55 and 16.6 to 22 GHz and 10 mm is 3.1 to 12.8 and 16.7 to 22 GHz	Higher than 1.69 dBi in 3–18 GHz range, 59% efficiency	Not much difference
[238]	CPW-fed circular monopole	34 × 25 × 0.135, ISM bands, ultra-wideband, WLAN band, WiMAX band, and 5G	PET, Ag NP	1.66–56.1 GHz (188.5%)	N/A	Bending side by side (yz plane) shifts the resonance point in the lower band, and the lower frequency of the bandwidth at a more moderate Bending in the top to bottom (xz plane) almost follows the same trend	More than 3 dB from 2.1- 56.1 GHz. Efficiency 85% to 99% from higher and lower frequency	N/A
[239]	Rectangular slotted metamaterial resonator patch antenna	60 × 60 × 2, IEEE 802.11 a and b/g/n WLAN, WiMAX, and GSM band	Low cost jeans, N/A	1.6–2.56 GHz (46%) and 4.24–7 GHz (49.11%)	2.45 GHz, 5.8 GHz.	Bandwidth and radiation patterns were affected due to bending. Below 30 mm bending radius, performance deteriorated much more.	1.6 dB in the lower band and 5 dB in the upper band.	N/A
[240]	Z-shaped microstrip patch antenna	45 × 36 × 0.135, dual-band Wi-Fi and wearable devices	PET, Ag NP	N/A,FBW = 23.33 % at 900 MHz and FBW = 11.66 % at 2.4 GHz.	900 MHz, 2.4 GHz.	The resonant frequency shifted from 0.9 GHz to 0.75 GHz and from 2.4 GHz to 2.2 GHz, and S11 changes −30.9 dB at 0.75 GHz and −34 dB at 2.2 GHz	16.74 and 16.24 dBi at 900 MHz and 2.4 GHz

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
