# Peer review of "Flexible Antennas: A Review"

_micromachines, 2020, doi:10.3390/mi11090847_

Round 1

Reviewer 1 Report

This manuscript reviews the materials, fabrication processes, and applications of flexible antennas.  However, the recent developments of flexible antennas (or even stretchable antennas) have been well provided and comprehensively reviewed.  The need for another review is not well outlined.  Please comment on any other reviews published on a similar topic and explicitly justify why there is room for another review.

While the authors have reviewed various materials and fabrication processes for flexible antennas with different application opportunities, the antenna performance parameters are less discussed and compared.  It is helpful to delineate the dependence of antenna performance on the conductivity of radiation elements, dielectric substrates, and different design considerations. The review paper is expected to go beyond the simple compilation of references.  The authors may want to include tables for comparison.  Additionally, please directly comment on the current importance of the field.

Author Response

Find attached the rebuttal letter

Reviewer 2 Report

In this paper, the authors present a quite thorough review of flexible antennas. While as a review paper, this paper has no novel technical contribution, it excels on completeness, clarity and depth. However, I believe it is important to make clear the position of this review and how it contributes to the body of knowledge. This is especially because there have been quite many reviews in this area and similar information could pretty much be obtained from those papers. Therefore, I believe the authors need to check thoroughly the list of references to make sure relevant review papers have been included. Most importantly they need to be able to justify accordingly how this review is different from those and explicitly describe in the introduction.

The authors also need to check the text thoroughly especially in terms of citation as I still found that some works have been referred wrongly, for instance, Fig. 22 and ref 229, Fig. 21 and ref 4, etc.

Lastly, please check the text thoroughly to avoid English errors.

Author Response

Find attached the rebuttal

Reviewer 3 Report

In this review manuscript, the authors focuses on the need for flexible antennas, materials and processes used for fabricating the antennas, various material properties influencing antenna performance, and specific biomedical applications accompanied by the design considerations. After a comprehensive treatment of the above mentioned topics, the authors focused in inherent challenges and future prospects of flexible antennas. Finally, an insight into the application of flexible antenna on future wireless solutions are discussed. I have to say this review manuscript is sufficient in terms of summarizing the advantages and disadvantages of antennas in flexible application. But I would like to give the authors minor modification below, and then it could be accepted.

  1. The annotations and icons in Figure 1 are not clear enough, and the relationship between each unit is a little confusing, the author needs to improve.
  2. The use of icon notes in figures 7 and 8 is unreasonable.
  3. The overall structure of individual parts in the article is unreasonable. For example:Part 4 “Special fabrication techniques for flexible wearable antennas” should be one of the parts of the part 3.
  4. The logic of application introduction is confusing. For example, in the article, parts 6, 7, 8 and 9 are all application introduction.
  5. The authors should supply more expectation of flexible antenna for future wireless solutions in part 11.

Author Response

Find attached the rebuttal file

Round 2

Reviewer 1 Report

Submission to a more specific journal is recommended. 

Reviewer 2 Report

Thank you for addressing the comments from the previous round. I believe the manuscript is suitable for a publication in this journal once the references are once again checked, e.g., missing ref 273, etc.